# X-Fi: A Modality-Invariant Foundation Model for Multimodal Human Sensing

**Xinyan Chen**,* **Jianfei Yang**[†]
Nanyang Technological University

## Abstract

Human sensing, which employs various sensors and advanced deep learning technologies to accurately capture and interpret human body information, has significantly impacted fields like public security and robotics. However, current human sensing primarily depends on modalities such as cameras and LiDAR, each of which has its own strengths and limitations. Furthermore, existing multimodal fusion solutions are typically designed for fixed modality combinations, requiring extensive retraining when modalities are added or removed for diverse scenarios. In this paper, we propose a modality-invariant foundation model for all modalities, X-Fi, to address these issues. X-Fi enables the independent or combinatory use of sensor modalities without additional training by utilizing a transformer structure to accommodate variable input sizes and incorporating a novel "X-fusion" mechanism to preserve modality-specific features during multimodal integration. This approach not only enhances adaptability but also facilitates the learning of complementary features across modalities. Extensive experiments conducted on the MM-Fi and XRF55 datasets, employing six distinct modalities, demonstrate that X-Fi achieves state-of-the-art performance in human pose estimation (HPE) and human activity recognition (HAR) tasks. The findings indicate that our proposed model can efficiently support a wide range of human sensing applications, ultimately contributing to the evolution of scalable, multimodal sensing technologies.

## 1 Introduction

Human sensing refers to using various sensors to capture human body information in a specific space, such as presence (Nguyen et al., 2016; Nanzer, 2017), activity (Vrigkas et al., 2015; Yang et al., 2023)), and pose (Toshev & Szegedy, 2014; Cai et al., 2022). Benefiting from advanced deep learning technology, human sensing has made significant progress in both accuracy and granularity, expanding applications in many domains. For instance, video surveillance (Elharrouss et al., 2021) is empowered by human sensing algorithms to analyze dangerous behaviors. In the field of robotics, human sensing helps robots comprehend human actions and realize human-robot collaboration (Gao et al., 2019).

Currently, human sensing tasks mainly rely on vision-based modalities like cameras (Ionescu et al., 2013), which face inherent limitations such as reliance on illumination, privacy concerns, and insufficient 3D information (Cai et al., 2022). Alternative modalities like LiDAR, mmWave radar, and WiFi have been introduced to overcome these challenges (Nirmal et al., 2021). Each modality has distinctive advantages but also has limitations, e.g., LiDAR offers high-resolution data but is expensive (Li et al., 2022), mmWava radar is cost-effective and informative but lacks sensitivity to static objects (An & Ogras, 2021), and WiFi is ubiquitous and privacy-preserving but has low resolution (Yang et al., 2023; Zhou et al., 2023). Therefore, single-modal sensing solutions cannot fit all scenarios, and a multi-modal approach that leverages the strengths of each modality is essential for future advancements in human sensing.

Numerous methods have been proposed for multi-modal perception based on sensor fusion. They usually predefine a fixed set of modalities for a specific scenario, e.g., combining RGB and LiDAR for pose estimation in autonomous driving to enhance long-range all-day sensing. Nevertheless, in any given scenario, once the model is trained, adding or removing even one modality requires

---

*Codes are available at: https://xyanchen.github.io/X-Fi

[†]Corresponding Author (jianfei.yang@ntu.edu.sg)

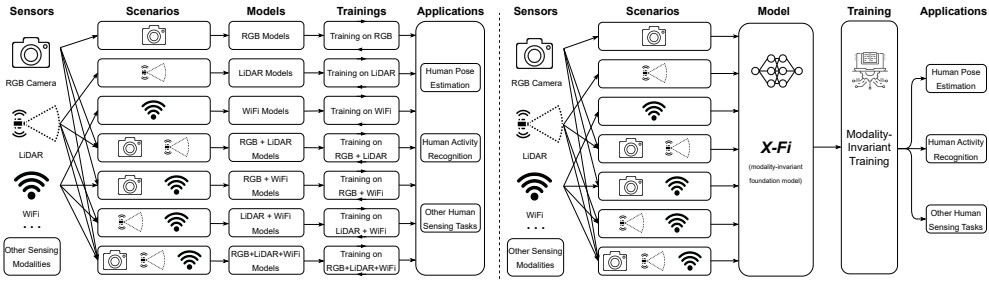

Figure 1: The left image depicts current human sensing solutions that are specifically designed and trained for fixed modality combinations, while the right image illustrates our proposed modality-invariant foundation model, X-Fi, which can be trained once and adapted to various scenarios.

a huge effort: adjusting the network and retraining it from scratch. In the real world, we may require versatile combinations of sensor modalities according to different scenarios (Yang et al., 2024). Hence, we contemplate whether it is possible to design **a one-for-all solution for modality-invariant human sensing**. Such a model would require training only once, allowing all sensor modalities that participated in the training process to be utilized independently or in any combination for a wide range of potential applications.

To achieve modality-invariant human sensing, we aim to design a foundation model that can dynamically handle varying numbers of modalities. First, considering the arbitrary number of input modalities, we naturally resort to a transformer structure due to its adaptability to variable input sizes (Vaswani et al., 2017). Second, recognizing that each modality has unique characteristics, we adopt modality-specific feature extractors to capture distinctive information. Finally, to retain each modality's distinct features during multimodal fusion, we introduce a cross-attention fusion mechanism that reduces the information loss caused by multimodal feature compression and fusion process. Surprisingly, this design not only supports dynamic inputs from various modalities but by training with diverse modality combinations to perform sensing tasks, it learns complementary features among modalities as well. As a result, it even achieves superior performance against conventional modality fusion methods such as feature-level and decision-level fusion.

To this end, we propose a novel modality-invariant foundation model, X-Fi, for versatile human sensing. X-Fi can take in any combination of modalities and activate corresponding parts to extract modality-specific features. A cross-modal transformer is designed to learn a cross-modal feature and each modality information is then preserved in the representation by executing cross-attention multimodal fusion processes to preserve distinctive modal features. As shown in the Figure 2, we have a set of modality-corresponded feature encoders and key-value generators to extract, fine-tune and preserve modality-specific features. For versatile combinations of sensor modalities, corresponding feature encoders are activated to obtain modality-specific features of each modality. Subsequently, the key-value generators are trained to produce key-value pairs, which contain more task-related information derived from modality-specific features. For the modality-invariant multimodal fusion process, we introduce "X-fusion" block. It consists of a cross-modal transformer and a set of modality-correlated cross-attention modules. The cross-modal transformer is designed to adaptively learn a unified cross-modal embedding across variable-sized multi-modal embedding, which comprises the features of each modality. The generated cross-modal embedding serves as the query, along with key-value pairs of each modality, are sent to the corresponding cross-attention module to perform attention-based feature injection. We evaluated X-Fi on human pose estimation (HPE) and human activity recognition (HAR) tasks on MM-Fi (Yang et al., 2024) and XRF55 (Wang et al., 2024) datasets, demonstrated that X-Fi surpasses previous methods by MPJPE 24.8% and PA-MPJPE 21.4% on HPE task, 2.8% on HAR task.

The main contributions are summarized as follows. First, we proposed X-Fi, a foundation model for modality invariant human sensing, which has strong scalability that only needs to be trained once and could support dynamic inputs from various modalities for a wide range of sensing scenarios. Second, we further developed X-Fusion block to empower efficient multi-modal fusion under modality invariant training strategy by preserving modality-specific information and applying token fusion technique to inject back into the cross-modal embedding. Third, we conducted extensive experiments on the MM-Fi and XRF55 datasets, utilizing a total of six distinct modalities, and observed that our proposed X-Fi achieved state-of-the-art performance on both the HPE and HAR tasks.

## 2 RELATED WORK

### 2.1 HUMAN SENSING

Human sensing has been extensively explored for applications such as autonomous driving and robotics in recent years, aiming to capture, analyze, and interpret human body information through various sensors and novel deep learning frameworks (Cai et al., 2022). Human sensing approaches are typically divided into three categories: i) Vision-based approaches, where human body information is extracted from visual data captured by cameras (Shahroudy et al., 2016) and leveraged by deep learning architectures like convolutional structures (Pavlakos et al., 2018; Sun et al., 2019; Li et al., 2020) and transformer (Goel et al., 2023; Cai et al., 2024). ii) Sensor-based approaches, using alternative modalities such as LiDAR (Ren et al., 2024), mmWave radar (Wang et al., 2021; An & Ogras, 2022), and WiFi CSI (Yang et al., 2023; Zhou et al., 2023) to overcome limitations of vision-based methods and expand applicability. iii) Multimodal fusion approaches, including decision-level fusion (An et al., 2022; Yang et al., 2024) and feature-level fusion (Zheng et al., 2022; Chen et al., 2023), which combine each modality's strengths to improve sensing performance.

### 2.2 MODALITY-INVARIANT METHODS

Current human sensing methods lack broad applicability. Feature-level fusion approaches (Zheng et al., 2022; Chen et al., 2023) are typically tailored to fixed modality combinations for specific applications. For example, (Chen et al., 2023) employed modality-specific backbones to extract grid features from images and cluster features from mmWave point clouds, combining them via a global integration module to produce fused features. Decision-level fusion approaches (Yang et al., 2024; Yadav et al., 2023) are easy to adapt to various modality combinations, but the performance could be hindered by less informative modalities, e.g. WiFi.

In fields beyond human sensing, several studies on multimodal fusion seek to accommodate diverse sensor inputs to a unified model for various applications. Some approaches apply cross-modal distillation techniques to seek the shared and the supplementary information across multi-modalities (Xue et al., 2022; Kong et al., 2019; Wang et al., 2020; Bruce et al., 2021). Some methods align multi-modalities features in a joint embedding space with a unified model structure (Man et al., 2023; Radford et al., 2021; Driess et al., 2023; Zhang et al., 2023; Girdhar et al., 2023; Moon et al., 2023; Tong et al., 2021; Wu et al., 2023). However, these approaches require retraining or fine-tuning to adapt to various scenarios. In recent works, foundation models, which are large-scale, pre-trained models that learn unified representations from diverse inputs, have gained attention for their capacity to be fine-tuned for a wide range of tasks (Devlin, 2018; Touvron et al., 2023). Extending this concept, modality-invariant representation learning have been proposed to handle dynamic multimodal combinations across various scenarios. (Xue & Marculescu, 2023) designs a gating network to dynamically activate the modality-combination-specific expert networks for multimodal fusion and (Memmel et al., 2023) introduces a modality-invariant training strategy by dropping modalities during training. However, these approaches face challenges due to the need for numerous networks and configurations to handle diverse modality combinations in the presence of extensive sensor options.

## 3 MODALITY-INVARIANT FOUNDATION MODEL FOR HUMAN SENSING

Assume the input $X$ comprises a random combination of modalities within a specific range, including any number and type of modalities. Our objective is to learn a unified representation $z_{cm}$ that integrates modality-specific features from $X$ by using a modality-invariant foundation model $\mathcal{F}$. This representation can be mapped to various down-stream human sensing task spaces $t_i$ by using corresponding task-specific functions $g_{t_i}$, such that the task-specified loss function $\mathcal{L}_{t_i}(g_{t_i}(\mathcal{F}(X)); y_{t_i})$ is minimized. In this paper, the $y_{t_i}$ refers to 3D human joints for HPE task and human activity classes for HAR task. Thus we propose X-Fi, a modality-invariant human sensing framework designed to map the arbitrary multimodal input $X$ to any task-specific target domain $t_i$.

As illustrated in the Figure 2, given $k$ modalities in the dataset, we first prepare a set of modality-specified feature encoders $\{E_i(\cdot; \theta_{E,i})\}_{i=1}^{k}$ to extract features $\{f_i \in \mathbb{R}^{n_f \times d_f}\}_{i=1}^{k}$ from corresponding modality inputs $\{x_i\}_{i=1}^{k}$, where $n_f$ indicates the number of features extracted for each modality and $d_f$ indicates the dimension of each feature. Then we introduce a cross-attention-based multimodal fusion design that could efficiently preserve modality-specific features in the learned unified

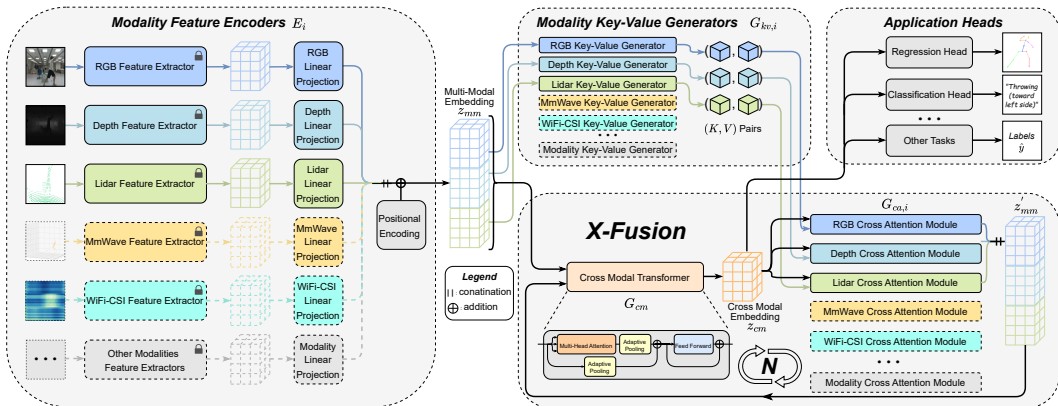

Figure 2: The architecture of the proposed modality-invariant foundation model, X-Fi. X-Fi consists modality feature encoders and an X-Fusion module, which includes a cross-modal transformer and modality-specified cross-attention modules. The modalities with dotted lines represent inactivate modalities in the given scenario. The $N$ in X-Fusion block represents the number of iterations.

cross-model embedding. The modality-specified key-value generators $\left\{ G_{kv,i} \left( \cdot ; \theta_{G,i}^{kv} \right) \right\}_{i=1}^{k}$ are applied to encode the modality information from $\{f_i\}_{i=1}^{k}$ into key-value pairs $\left\{ (K_i, V_i) \right\}_{i=1}^{k}$. To obtain an unified cross-modal embedding $z_{cm} \in \mathbb{R}^{n_f \times d_f}$, we use a cross-modal transformer $G_{cm} \left( \cdot ; \theta_G^{cm} \right)$ to learn from the $z_{mm} \in \mathbb{R}^{(n_f * k) \times d_f}$. The generated cross modal embedding $z_{cm}$ serves as the query $Q$, together with each key-value pair, to send into the modality-specific cross-attention modules $\left\{ G_{ca,i} \left( \cdot ; \theta_{G,i}^{ca} \right) \right\}_{i=1}^{k}$ in order to inject modality-specific information. Through iterative learning of the $z_{cm}$ and performing modality-specific information injection, the final produced $z_{cm}$ could be used for various downstream tasks via task-correlated Multi-layer Perception (MLP) head.

In the following subsections, we will decompose X-Fi by describing the modality feature encoding (sec. 3.1), the attention-based X-Fusion (sec. 3.2), and the modality-invariant training (sec. 3.3).

## 3.1 MODALITY FEATURE ENCODING

In human sensing, different modalities exhibit distinctive characteristics, requiring unique feature encoder structures to extract robust modality-specific feature. For instance, images are represented as a 2D matrices and have rich spatial features, requiring models like CNN (LeCun et al., 1998) and ViT (Dosovitskiy, 2020) to extract vision features. While point clouds from LiDAR consist of a set of unordered points in Euclidean space, necessitating point cloud-specific frameworks like Point Net (Qi et al., 2017) and Point Transformer (Zhao et al., 2021) to process them. Thus, we introduce modality-specific feature encoders in X-Fi to extract features from various modalities.

In X-Fi, the modality-specified feature encoder $E_i \left( \cdot ; \theta_{E,i} \right)$ for each modality consists of a pretrained feature extractor $F_i \left( \cdot ; \theta_{F,i} \right)$ and a linear projection head. The feature extractors are responsible for extracting informative high-dimensional representations from modal data. To accommodate diverse combinations of sensor modalities across different application scenarios, we can select any model structures as backbones for these feature extractors. Each feature extractor is modality specified, which allows us to choose a model structure that best aligns with the unique characteristics of the modality's data, thereby ensuring the efficiency of feature extraction. For example: for point clouds from mmWave radar and LiDAR, we adopt the Point Transformer backbone (Zhao et al., 2021). The self-attention mechanism in Point Transformers is particularly suitable for point cloud data due to its permutation invariance and ability to dynamically focus on relevant points.

Since the data structures of the features extracted by different modality-specific feature extractors are still different across modalities, we use a set of linear projection structures to unify these features into the same dimensional space, allowing them to be used together in subsequent stages of the model. Finally, positional encoding is added to obtain the final features.

$$\{E_i(\cdot;\theta_{E,i})\}_{i=1}^{k} = \{\text{LinearProjection}_i(F_i(\cdot;\theta_{F,i});\theta_{LP,i})\}_{i=1}^{k} \quad (1)$$

## 3.2 ATTENTION-BASED X-FUSION

After obtaining the modality-specific features, a natural question is posed: How can the modality-invariant foundation model effectively combine these features to obtain a unified cross-modal representation that is applicable for downstream human sensing tasks? Thus, in this section, we will interpret the proposed X-Fusion structure which consists of a cross-modal transformer to learn unified cross-modal features and cross-attention fusion modules to preserve distinctive modality features.

### 3.2.1 CROSS-MODAL TRANSFORMER

In this subsection, we aim to learn a unified cross-modal feature. The proposed structure should be capable of dynamically handling varying numbers of modality features and effectively achieving information interaction among diverse modal features. Thus, we adopt a Transformer-based structure (Vaswani et al., 2017) that can accept variable sizes of input. Also, the self-attention mechanism can equally learn the long-distance dependencies between any two modal features in the input, thereby effectively modeling global information for all modal features.

First, we concatenate all obtained modality-specific features to initialize a multi-modal embedding $z_{mm} \in \mathbb{R}^{(n_f * k) \times d_f}$. As illustrated in the Figure 2, we then linearly map the multi-modal embedding into three vectors, named $Q_{mm}, K_{mm}, V_{mm}$. Next, we perform dot-product-based multi-head attention to calculate the attention value which models the dependencies among modality features.

Notably, the multi-head attention and feed-forward blocks output embedding of the same size as the input, resulting in variable-sized cross-modal embedding based on the number of modality inputs. This variability complicates the design of subsequent fusion blocks and task-specific projection heads which have fixed-size parameters. To address this issue, we apply adaptive pooling both after the multi-head attention block and within the residual connection to perform downsampling and obtain unified-sized outputs $o_{mm} \in \mathbb{R}^{n_f \times d_f}$. Finally, we feed $o_{mm}$ into a feed-forward structure with a residual connection to obtain the unified cross-modal embedding $z_{cm} \in \mathbb{R}^{n_f \times d_f}$.

$$o_{mm} = \text{AvgPool}(\text{MultiHead}\,(Q_{mm}, K_{mm}, V_{mm})) + \text{AvgPool}(z_{mm}) \tag{2}$$

$$z_{cm} = FFN(o_{mm}) + o_{mm} \tag{3}$$

The adaptive average pooling performs the dimension reduction for the multimodal feature and the sequential feature, which are summed up for the following layer. Then the random modality dropping in training ensures that all possible combinations of modalities are learned by X-Fi, implicitly aligning the feature of each modality. Therefore the final feature can be modality-invariant, demonstrated by massive empirical studies using 38 modality combinations on two large-scale datasets.

### 3.2.2 CROSS-ATTENTION MULTI-MODAL FUSION

However, the basic transformer design, which has a layered architecture with sequentially stacked transformer blocks, often prioritize certain modalities during training and the inference performance will rely on the existence of the dominant modalities. The reason is that the self-attention mechanism in the cross-modal module tends to focus on the more intuitive and informative modality-specific features during training (Lu et al., 2019). Such focus results in significant information loss from less dominant modalities, compromising robust and equitable performance across different scenarios where dominant modalities may be distorted or missing (Ma et al., 2022; Memmel et al., 2023). To address this, we propose cross-attention fusion modules that reintegrate overlooked modality-specific information into the cross-modal embeddings, thereby encouraging effective learning from less-informative modalities during training. Rather than impeding the features extraction from more informative modalities, such approach enhances the model's ability to capture complementary features across modalities, improving robustness across diverse application scenarios. We also design an X-Fusion process that iterates on the same block, compelling the X-Fusion block to identify a unified dynamic attention allocation method that effectively preserves the features of each modality and optimally integrates them to achieve the most optimal solution.

**Key-Value Generators.** To prepare key-value pairs that preserve modality-specific information, we design key-value generators for each modality. As illustrated in the Figure 7, key-value generator $G_{kv,i}$ receives modality specific feature $f_i \in \mathbb{R}^{n_f \times d_f}$ extracted from feature encoder $E_i$ as the input. $f_i$ is first passed through a two-layer MLP with a hidden dimension of $d_f * 2$ incorporating a Rectified Linear Unit (ReLU) activation (Nair & Hinton, 2010) and layer normalization in between. The MLP

output is then projected to key $K_i$ and value $V_i$ using separate linear mapping layers. Once the key-value pair $(K_i, V_i)$ is generated from $f_i$, it remains unchanged during iterations in the X-fusion block to prevent information leakage between modalities.

**Cross-Attention Modules.** We employ a series of cross-attention modules to integrate modality-specific information into the cross-modal embedding $z_{cm}$. Each transformer is associated with a specific modality and receives three inputs: the unified cross-modal embedding $z_{cm}$ that is shared across all cross-attention modules, the key $(K)$ and value $(V)$ pairs obtained from modality-specific key-value generators. Take the $i$-th modality as an example, as illustrated in the Figure 7, $z_{cm}$ serves as the query $(Q)$ in the multi-head attention mechanism. Attention weights are computed based on $Q$ and $K_i$ to scale the modality-specific value $V_i$, yielding an intermediate result denoted as $o_{ca,i}$. Subsequently, $z_{cm}$ is residual added to $o_{ca,i}$ to retain cross-modal features, resulting in $o'_{ca,i}$. The final output $f'_i$ of each modality-specific cross-attention module is produced by passing $o'_{ca,i}$ through a feed-forward structure with a residual connection. Each $f'_i$ generated from the modality-specific cross-attention module encapsulates fused information focusing on a single modality. To integrate information from all modalities, we concatenate all $\{f'_i\}_{i=1}^k$ to get the updated multi-modal embedding $z'_{mm}$, which is used in the subsequent X-fusion block iteration.

$$z'_{mm} = \text{Concat}\big\{ G_{ca,i}\left( z_{cm}, K_i, V_i; \theta_{G,i}^{ca} \right) \big\}_{i=1}^k \qquad (4)$$

### 3.3 MODALITY-INVARIANT TRAINING

To train our modality invariant foundation model, we design an explicit training strategy with reference to (Memmel et al., 2023). First, to ensure that the feature extractors retain their ability to capture modality-specific features without being biased, we pre-train the parameters on the respective datasets and freeze these parameters during subsequent training. Next, to simulate versatile combinations of sensor modalities in different scenarios, we use a modality existence list containing $n$ elements to indicate whether each modality is present in the input modal combination. This modality existence list is utilized to control the presence of input modalities during each training iteration and activate the corresponding modality-specific structures in the X-Fi. In addition, each modality is assigned an independent occurrence probability $p_i$. By adjusting the modality-corresponded $p_i$, we can refine the model's training strategy by prioritizing certain modalities. For example, because LiDAR point cloud data is inherently sparser than image data and exhibits complex features such as permutation invariance, it is necessary to include more LiDAR samples in the training process to ensure accurate model fitting. Since the training sample is controlled by the modality existence list, we can model the training samples by the distribution of the modality existence list. We denote the count of occurrences of $i$-th modality in $m$ training iterations by a variable $K_i$, and $K_i \sim Binomial(m, p_i)$. Since $K_i$ for each modality is independent, we can describe the distribution using a joint probability mass function $P(K_1 = k_1, \cdots, K_n = k_n) = \prod_{i=1}^n \binom{m}{k_i} p_i^{k_i} (1-p_i)^{m-k_i}$. For each training batch, the modality existence list is sampled from a distribution to regulate the input modalities, thereby influencing the activation of modality-specific model structures.

## 4 EXPERIMENTS

### 4.1 EXPERIMENTAL SETUP

**Datasets.** We train and evaluate our proposed X-Fi on the two largest human sensing multimodal public datasets, MM-Fi (Yang et al., 2024) and XRF55 (Wang et al., 2024), to assess its efficiency as a unified modality-invariant foundation model across diverse human sensing tasks, including Human Pose Estimation (HPE) and Human Activity Recognition (HAR). The dataset details for each task are as follows: (i) HPE: MM-Fi consists of over 320k synchronized frames from 40 human subjects. Each synchronized frame consists of samples collected from 5 sensing modalities, including RGB images (I), Depth images (D), LiDAR point clouds (L), mmWave point clouds (R), and WiFi-CSI (W). The chosen annotation type is 3D whole-body key points, representing the positions of 17 human joints. (ii) HAR: MM-Fi and XRF55 are utilized for HAR task evaluation. MM-Fi consists of 27 action categories, including 14 daily activities and 13 rehabilitation exercises. We exclude WiFi-CSI modality from MM-Fi when addressing the HAR task, as each WiFi data sample spans only 100ms, which is insufficient to effectively capture the distinct data patterns associated with different actions. XRF55 includes 42.9K radio frequency samples collected by 3 sensing modalities,

| Metrics | MPJPE | | | | | PA-MPJPE | | | | |
|---|---|---|---|---|---|---|---|---|---|---|
| MM-Fi | X-Fi | Baseline1 | Imp↑ | Baseline2 | Imp↑ | X-Fi | Baseline1 | Imp↑ | Baseline2 | Imp↑ |
| I | 93.9 | 121.0 | 22.4% | 121.0 | 22.4% | 60.3 | 68.0 | 11.4% | 68.0 | 11.4% |
| D | 101.8 | 102.4 | 0.6% | 102.4 | 0.6% | 48.4 | 52.7 | 8.2% | 52.7 | 8.2% |
| L | 167.1 | 161.5 | -3.5% | 161.5 | -3.5% | 103.2 | 103.5 | 0.3% | 103.5 | 0.3% |
| R | 127.4 | 141.3 | 9.8% | 141.3 | 9.8% | 69.8 | 72.4 | 3.7% | 72.4 | 3.7% |
| W | 225.6 | 227.1 | 0.7% | 227.1 | 0.7% | 105.3 | 108.0 | 2.5% | 108.0 | 2.5% |
| I+D | 86.1 | 96.7 | 11.0% | 109.2 | 21.2% | 48.1 | 53.9 | 10.7% | 57.1 | 15.6% |
| I+L | 93.0 | 122.1 | 23.8% | 121.2 | 23.3% | 59.7 | 75.6 | 21.1% | 73.0 | 18.2% |
| I+R | 88.8 | 101.0 | 12.1% | 140.6 | 36.8% | 57.3 | 57.3 | 0.0% | 68.2 | 16.0% |
| I+W | 93.0 | 146.4 | 36.5% | 156.7 | 40.7% | 59.5 | 77.3 | 23.1% | 86.3 | 31.1% |
| D+L | 102.5 | 111.7 | 8.2% | 108.0 | 5.1% | 48.4 | 68.8 | 29.7% | 55.8 | 13.3% |
| D+R | 98.0 | 94.1 | -4.1% | 116.4 | 15.8% | 47.3 | 51.8 | 8.6% | 57.0 | 17.0% |
| D+W | 101.8 | 141.7 | 28.1% | 155.5 | 34.5% | 48.1 | 71.4 | 32.6% | 81.5 | 41.0% |
| L+R | 109.8 | 116.4 | 5.7% | 137.0 | 19.9% | 63.4 | 71.5 | 11.3% | 71.9 | 11.8% |
| L+W | 159.5 | 167.1 | 4.5% | 206.2 | 22.6% | 102.7 | 100.7 | -2.0% | 109.3 | 6.0% |
| R+W | 117.2 | 144.2 | 18.7% | 141.3 | 17.0% | 62.7 | 72.2 | 13.0% | 77.6 | 19.1% |
| I+D+L | 84.8 | 102.9 | 17.5% | 104.5 | 18.8% | 48.2 | 62.3 | 22.7% | 61.2 | 21.4% |
| I+D+R | 83.4 | 86.7 | 3.8% | 113.4 | 26.5% | 47.3 | 50.1 | 5.4% | 58.0 | 18.4% |
| I+D+W | 85.3 | 118.6 | 28.0% | 137.8 | 38.1% | 48.1 | 63.7 | 24.4% | 75.3 | 36.1% |
| I+L+R | 88.4 | 101.4 | 12.8% | 135.0 | 34.5% | 57.2 | 62.7 | 8.8% | 67.7 | 15.6% |
| I+L+W | 93.0 | 134.9 | 31.1% | 156.2 | 40.5% | 59.7 | 82.0 | 27.2% | 84.8 | 29.6% |
| I+R+W | 88.5 | 116.5 | 24.1% | 128.9 | 31.3% | 57.1 | 63.1 | 9.5% | 75.0 | 23.9% |
| D+L+R | 96.0 | 95.3 | -0.8% | 111.3 | 13.7% | 47.3 | 58.6 | 19.2% | 58.4 | 19.0% |
| D+L+W | 102.0 | 130.7 | 22.0% | 154.6 | 34.0% | 48.1 | 78.1 | 38.4% | 84.1 | 42.8% |
| D+R+W | 97.0 | 113.2 | 14.3% | 132.0 | 26.5% | 47.1 | 59.4 | 20.7% | 71.8 | 34.3% |
| L+R+W | 107.4 | 129.6 | 17.2% | 141.6 | 24.1% | 63.1 | 78.1 | 19.2% | 77.0 | 18.1% |
| I+D+L+R | 83.5 | 90.7 | 7.9% | 109.4 | 23.6% | 47.6 | 55.8 | 14.6% | 61.6 | 22.8% |
| I+D+L+W | 86.0 | 116.7 | 26.4% | 134.6 | 36.2% | 48.2 | 70.0 | 31.2% | 75.4 | 36.0% |
| I+D+R+W | 84.0 | 101.9 | 17.6% | 128.6 | 34.7% | 47.6 | 56.3 | 15.5% | 71.3 | 33.3% |
| I+L+R+W | 88.6 | 114.0 | 22.3% | 174.6 | 49.2% | 57.1 | 69.5 | 17.8% | 78.1 | 26.8% |
| D+L+R+W | 97.6 | 110.8 | 11.9% | 147.6 | 33.9% | 47.4 | 66.7 | 28.9% | 73.1 | 35.1% |
| I+D+L+R+W | 83.7 | 103.0 | 18.8% | 128.9 | 35.1% | 47.6 | 62.4 | 23.8% | 72.5 | 34.3% |

Table 1: Performance comparisons of X-Fi with baseline methods on the MM-Fi dataset for HPE task. "Baseline1" denotes the decision-level fusion results and "Baseline2" denotes the feature-level fusion results. Imp denotes the improvement achieved over baseline in percentage level.

including mmWave Range-Doppler & Range-Angle Heatmaps (R), WiFi-CSI (W), and RFID phase series data (RF). XRF55 comprises 55 action classes collected from 39 human subjects, covering categories such as fitness activities and human-computer interactions. We follow the S1 Random Split for MM-Fi and the original split setting for XRF55 as outlined in their respective papers.

**Training Objective.** Different loss functions are applied as the training objectives for different human sensing tasks. For HPE, Mean-Squared-Error is chosen as the loss function, which measures the average squared euclidean distance between each of the 17 predicted whole-body key points and their corresponding ground truth points in 3D space. For HAR, Cross-Entropy loss is applied between the predicted probability distribution and the ground truth distribution.

**Baseline Formulation.** To evaluate the performance of our proposed unified modality-invariant foundation model across various human sensing tasks and modality combinations, we formulate baseline models by adopting a feature fusion approach (Das et al., 2021; Marcard et al., 2016) and a decision-level fusion approach (Yang et al., 2024). For the feature fusion approach, X-Fi uses pretrained modality-specific extractors to obtain modality features. Single-modal inputs are projected into the task space via an MLP, while multi-modal inputs are concatenated and fused using an MLP with average pooling to map them into the unified task representation space. The baseline models are retrained on each modality combination separately. For different tasks, we retrain baseline models' weights (including the pre-trained feature extractor) using different learning objectives. For the decision-level fusion approach, we first obtain the final-layer output of each individual modality. Then, we calculate the average of the outputs from all modalities in the combination to generate the fused output. The baseline results are presented in the Table 1 & 2.

**Implementation Details.** We conduct experiments on all human sensing tasks and datasets with the following model settings. To standardize feature representations obtained by various modality-specific feature extractors, we apply linear projection units to map each modality feature representation to $n_f = 32$, each with a feature dimension of $d_f = 512$. The positional embedding, derived from LiDAR 3D point cloud raw data due to its inherent capture of human body spatial information,

is only provided in the multi-modal embedding if the LiDAR modality is present in the input. The backbone for both the cross-modal transformer and each modality-specific cross-attention module consists of a 1-layer decoder-only transformer structure with 8 multi-head attention heads and a scaling factor of 0.125. Each modality-specific key-value pair and the cross-modal embedding in X-Fusion block comprise 32 features with a feature dimension of 512. The number of iterations on the X-Fusion block is set to a default of 4 in our experiments. The AdamW optimizer, with an initial learning rate of $1 \times e^{-3}$ for HPE and $1 \times e^{-4}$ for HAR, is chosen for model optimization. The training process is performed with a batch size of 16 on an NVIDIA GeForce RTX 4090 GPU.

## 4.2 QUANTITATIVE RESULTS

We compare our approach to baseline methods in downstream human sensing tasks, including HPE and HAR. X-Fi results are obtained by evaluating a single X-Fi model, only trained once using the proposed modality-invariant training strategy, across various modality combinations during testing. And the baseline results are retrained on each modality combination separately. For convenience, we refer to the decision-level fusion baseline as "baseline1", and the feature-level fusion baseline as "baseline2". The results for X-Fi, as shown in Tables 1 & 2, represent average values obtained from multiple experiments.

**Human Pose Estimation.** We assess the performance of X-Fi on the MM-Fi dataset (Yang et al., 2024) by adopting two commonly-used metrics in HPE: Mean Per Joint Position Error (MPJPE) and Procrustes Analysis MPJPE (PA-MPJPE). For single-modal inputs, X-Fi improves the averaged MPJPE and PA-MPJPE by 6.0% and 5.2%, respectively. For the dual-modality inputs, the averaged MPJPE and PA-MPJPE are improved respectively by 14.5% and 14.8% on baseline1, 23.7% and 18.9% on baseline2. Notably, for LiDAR and WiFi modalities, when used as individual inputs, X-Fi does not achieve good performance, with MPJPE difference of $-3.5\%$ and $+0.7\%$, respectively. However, when the two modalities are combined, the performance improves significantly, surpassing the baseline1 by

Table 2: HAR accuracy (%) on the MM-Fi and XRF55 datasets.

| Modality | Baseline | X-Fi | Imp ↑ |
|---|---|---|---|
| MM-Fi | | | |
| I | 25.3 | 26.5 | 1.2% |
| D | 49.9 | 48.1 | -1.8% |
| L | 63.9 | 52.7 | -11.2% |
| R | 85.0 | 85.7 | 0.7% |
| I + D | 45.2 | 45.3 | 0.1% |
| I + L | 23.8 | 35.2 | 11.4% |
| I + R | 66.9 | 73.4 | 6.6% |
| D + L | 49.7 | 51.6 | 1.9% |
| D + R | 74.1 | 79.8 | 5.7% |
| L + R | 85.6 | 88.7 | 3.2% |
| I + D + L | 43.0 | 48.7 | 5.7% |
| I + D + R | 69.1 | 70.7 | 1.6% |
| I + L + R | 68.2 | 77.8 | 9.6% |
| D + L + R | 72.9 | 80.5 | 7.6% |
| I+D+L+R | 72.6 | 72.2 | -0.4% |
| XRF55 | | | |
| R | 82.1 | 83.9 | 1.9% |
| W | 77.8 | 55.7 | -22.1% |
| RF | 42.2 | 42.5 | 0.3% |
| R+W | 86.8 | 88.2 | 1.4% |
| R+RF | 71.4 | 86.5 | 15.1% |
| W+RF | 55.6 | 58.1 | 2.6% |
| R+W+RF | 70.6 | 89.8 | 19.2% |

4.5% and the baseline2 by 22.6%. For the triple-modality inputs, X-Fi demonstrates an average improvement of 17.0% and 28.8% in MPJPE, 19.5% and 25.9% in PA-MPJPE. For the combined input of four modalities, the results indicate an average enhancement of 17.0% and 35.5% in MPJPE, 21.6% and 30.8% in PA-MPJPE. For all five modalities combined input, MPJPE increases by 18.8% and 35.1%, PA-MPJPE increases by 23.8% and 34.3%.

**Human Activity Recognition.** The performance of X-Fi on the HAR task is evaluated by classification accuracy on both the MM-Fi (Yang et al., 2024) and XRF55 (Wang et al., 2024) datasets. On the MM-Fi dataset, X-Fi performs better than the baseline when handling multimodal inputs, yielding an average improvement of 2.8%. But when dealing with single-modality inputs, its performance declines, dropping by 2.8%. On the XRF55 dataset, X-Fi only encounters challenges with WiFi-CSI single-modality input. It outperforms the baseline on all other modality combinations, achieving an average improvement of 6.7%. Notably, while X-Fi's performance is suboptimal when using WiFi as a single input, when combined with other modalities, WiFi not only avoids hindering performance but also introduces unique information that enhances classification accuracy. For example, when X-Fi processes mmWave radar and RFID multimodal input, it outperforms the baseline by 15.1%. With the inclusion of WiFi, this improvement increases to 19.2%.

## 4.3 ANALYTICS

**Impact of modality existence list.** To assess the impact of adjusting the independent existing probability of each modality in the modality existence list, we conduct experiments with different probability combinations on the XRF55 dataset for the HAR task. The XRF55 modality existence list contains the independent existence probabilities of mmWave Radar, WiFi-CSI, and RFID modalities in sequential order. As shown in Table 3, X-Fi encounters difficulties in dealing with WiFi-CSI. However, through gradually increasing the existing probability of WiFi-CSI, the model is priori-

tized to fit WiFi-CSI modality during training, which yields significant improvement from $29.1\%$ to $55.7\%$ in handling single WiFi-CSI input. Additionally, we are surprised to find that prioritizing a particular modality does not degrade the model's ability to extract features from the other two modalities. Consequently, the multimodal fusion results are also improved a lot.

| XRF55 | Accuracy (%) | | |
| --- | --- | --- | --- |
| $(P_R,P_W,P_{RF})$ | (0.5,0.5,0.8) | (0.5,0.7,0.7) | (0.5,0.9,0.6) |
| R | 82.2 | 82.4 | **83.9** |
| W | 29.1 | 37.4 | **55.7** |
| RF | 41.6 | 41.9 | **42.5** |
| R+W | 85.1 | 86.0 | **88.2** |
| R+RF | 84.5 | 84.8 | **86.5** |
| W+RF | 50.3 | 52.7 | **58.1** |
| R+W+RF | 86.4 | 87.3 | **89.8** |

Table 3: Comparisons of X-Fi trained with different modality existence lists. The probabilities are shown in decimal form.

| MM-Fi | MPJPE (mm) | | PA-MPJPE (mm) | |
| --- | --- | --- | --- | --- |
| Method | VO Transformer | X-Fi | VO Transformer | X-Fi |
| I | 99.6 | **93.9** | 65.0 | **60.3** |
| L | 212.2 | **167.1** | 109.6 | **103.2** |
| R | 138.0 | **127.4** | 74.6 | **69.8** |
| W | 243.3 | **225.6** | 112.9 | **105.3** |
| L + R | 123.4 | **109.8** | 69.9 | **63.4** |
| L + W | 216.3 | **159.5** | 112.7 | **102.7** |
| I + L + W | 96.6 | **93.0** | 61.5 | **59.7** |

Table 4: Comparison between the SOTA method (Visual Odometry Transformer) and X-Fi on the HPE task.

**Comparison between existing SOTA (Visual Odometry Transformer) and X-Fi.** To demonstrate the efficiency of X-Fi on modality-invariant human sensing tasks, we conduct comparison with existing SOTA method (Memmel et al., 2023) that utilize basic transformer encoder consisting of four stacked blocks to performance modality fusion. A subset of comparison results are analyzed in Table 4 and the full findings are provided in the appendix 5. As shown in Table 4, X-Fusion yields better performance on most of the modality combination input. First, when handling single modality input, transformer-based fusion struggles with challenging modalities, whereas X-Fusion effectively captures the distinctive features of each modality. For example, performance with LiDAR input improves by $21.3\%$. Next, in contrast to transformer-based fusion methods, X-Fusion leverages each modality's strengths, yielding better fusion results than single-modality inputs. X-Fusion demonstrates superior performance compared to either the LiDAR or WiFi-CSI modality when used individually. Lastly, in the case of multi-modality inputs, the performance of transformer-based fusion is highly dependent on the presence of dominant modalities. For example, adding the RGB modality improves the performance of the LiDAR and WiFi-CSI multi-modal inputs by 119.7 mm with the transformer backbone, whereas the corresponding improvement with X-Fusion is 66.6 mm.

**Comparison between X-Fusion and its variants.** To assess the effectiveness of the X-Fusion structure, which iterates multiple loops within the same block for multimodal feature fusion, we design two variants. The first employs a layered architecture with four sequentially stacked X-Fusion blocks, where each layer learns new key-value pairs from the multimodal embedding of the previous layer. The second variant is similar but uses the same key-value pairs across layers, learned from the embedding generated by the modality feature encoder. We conduct experiments on both variants, with full results provided in the appendix 5. On average, the first variant demonstrates a performance improvement of +1.27 mm for MPJPE and -0.7 mm for PA-MPJPE across all modality combinations. The second variant achieves an improvement of 0.93 mm for MPJPE and 0.03 mm for PA-MPJPE. Considering the computational efficiency and the minor performance differences, we retain the X-Fusion structure where an X-Fusion block is applied iteratively.

### 4.4 QUALITATIVE RESULTS

To intuitively illustrate the performance of X-Fi on the human sensing tasks, we visualize the estimated human skeleton generated by the model in the HPE task in the Figure 3 and employ the t-SNE method (Van der Maaten & Hinton, 2008) to visualize the multimodal embedding obtained by the X-fusion block for the HAR task in the Figure 4.

**Human Pose Estimation.** The visualization results for HPE comprise two actions, 'picking up things' and 'throwing', each depicted through a sequence of four images. In different rows, we presented the results for the RGB and depth single-modal inputs, the fused results of the RGB and depth multimodal inputs, and the ground truth. To facilitate a clearer comparison between the fused results and the single-modal inputs results, we incorporated blue and orange dashed lines in the fused result images to represent the results of the RGB and depth single-modality inputs, respectively. Through observation, we found that the RGB modality provides more accurate estimations of leg movements, while the depth modality achieves greater accuracy in estimating hand movements. By combining both modalities into a multimodal input, X-Fi effectively retains the leg movement accuracy from the RGB modality and the hand movement accuracy from the depth modality, resulting in fused outcomes that are closest to the ground truth.

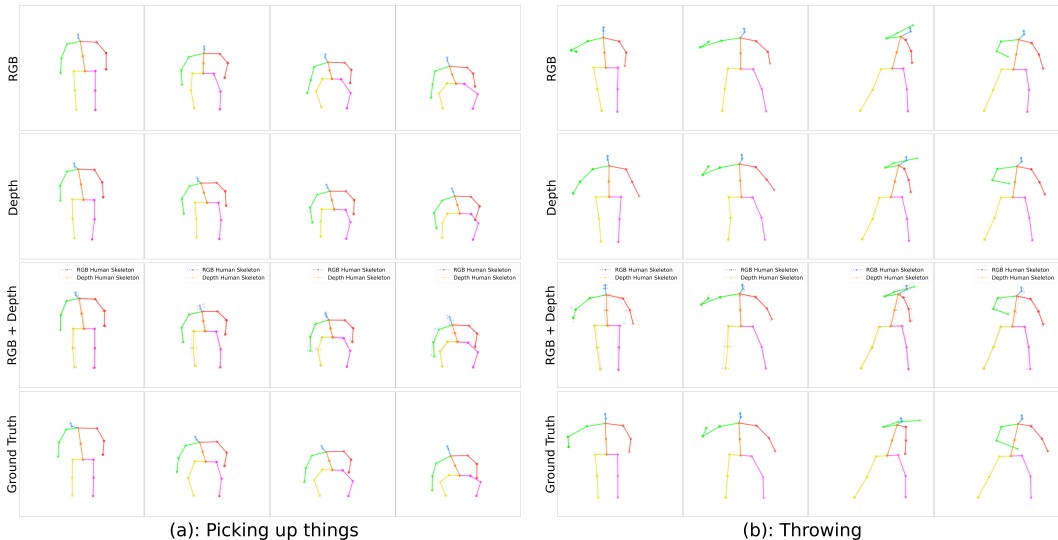

(a): Picking up things
(b): Throwing

Figure 3: Comparison of predicted human skeletons.

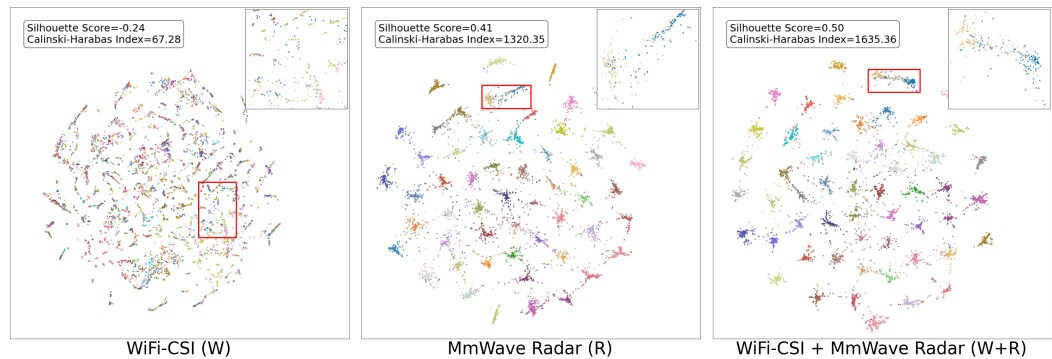

WiFi-CSI (W)          MmWave Radar (R)          WiFi-CSI + MmWave Radar (W+R)

Figure 4: Comparison of multi-modal embedding distribution for HAR. The upper right corner of the image provides an enlarged view of the red-boxed area from the original image.

**Human Activity Recognition.** The visualization results for HAR include the multimodal embedding distributions from WiFi-CSI and mmWave Radar single-modal inputs, as well as the combined WiFi-CSI and mmWave Radar multi-modal inputs. To more closely analyze the distribution of sample points, we zoomed in on a small region containing points from two distinct categories. To quantify the distribution, we used the Silhouette score and Calinski–Harabasz index as indicators of clustering quality. The Silhouette score assesses how well an object fits within its own cluster compared to others, ranging from -1 to 1, with higher values indicating better separation and cohesion. Similarly, higher Calinski–Harabasz index values reflect more distinct clusters. We observed that the multimodal input results exhibit superior feature discriminability. While the mmWave Radar single-modal results demonstrate good classification capability, a closer inspection of the zoomed-in region reveals that the multimodal input results form tighter clusters. Furthermore, a comparison of the indicator values confirms that the multimodal results outperform any single-modal results.

## 5    CONCLUSION

In this paper, we introduce X-Fi, a modality-invariant foundation model designed to overcome the limitations of current human sensing approaches, which often rely on fixed modality combinations. By leveraging a transformer architecture and a novel X-fusion mechanism, X-Fi supports flexible multimodal integration and effectively utilizes distinctive modal information. Our extensive experiments on the MM-Fi and XRF55 datasets demonstrate the model's superior performance in both human pose estimation (HPE) and human activity recognition (HAR) tasks. These results highlight X-Fi's potential to drive advancements in scalable, multimodal human sensing technologies.

**Acknowledgement.** This work is supported by a Start-up Grant from Nanyang Technological University and jointly funded by the Singapore MOE Tier-1 research grant.

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

# A APPENDIX

| MM-Fi | MPJPE | | | | PA-MPJPE | | | |
|---|---|---|---|---|---|---|---|---|
| Methods | Visual Odometry Transformer | X-Fusion (iterative) | X-Fusion (stacked) | X-Fusion (stacked, same kv) | Visual Odometry Transformer | X-Fusion (iterative) | X-Fusion (stacked) | X-Fusion (stacked, same kv) |
| I | 99.6 | **93.9** | 100.6 | 95.8 | 65.0 | **60.3** | 62.2 | 61.3 |
| D | 109.0 | 101.8 | **94.8** | 104.5 | 56.1 | **48.4** | 50.8 | 48.8 |
| L | 212.2 | 167.1 | 184.9 | **160.2** | 109.6 | **103.2** | 104.7 | 103.6 |
| R | 138.0 | 127.4 | **121.2** | 122.8 | 74.6 | 69.8 | **65.4** | 66.4 |
| W | 243.3 | 225.6 | 229.7 | **224.3** | 112.9 | 105.3 | 105.4 | **103.2** |
| I + D | 89.7 | 86.1 | **81.1** | 84.9 | 52.9 | **48.1** | 50.0 | **48.1** |
| I + L | 96.6 | **93.0** | 98.8 | 93.2 | 62.4 | **59.7** | 60.9 | 60.2 |
| I + R | 93.0 | **88.8** | 91.9 | 89.8 | 59.1 | **57.3** | 57.5 | 58.3 |
| I + W | 96.0 | **93.0** | 97.2 | 94.0 | 61.9 | **59.5** | 60.9 | 60.5 |
| D + L | 97.3 | 102.5 | **91.4** | 97.9 | 52.3 | 48.4 | 49.5 | **48.0** |
| D + R | 94.9 | 98.0 | **88.4** | 96.8 | 50.6 | 47.3 | 48.0 | **47.3** |
| D + W | 95.6 | 101.8 | **94.5** | 98.8 | 51.4 | 48.1 | 49.6 | **48.0** |
| L + R | 123.4 | 109.8 | 115.5 | **106.9** | 69.9 | 63.4 | 64.1 | **61.7** |
| L + W | 216.3 | 159.5 | 181.7 | **156.4** | 112.7 | 102.7 | 103.9 | **102.5** |
| R + W | 126.1 | **117.2** | 120.8 | 117.4 | 70.0 | **62.7** | 63.9 | **62.7** |
| I + D + L | 86.8 | 84.8 | **78.6** | 84.2 | 52.2 | 48.2 | 49.3 | **48.0** |
| I + D + R | 85.2 | 83.4 | **78.2** | 83.8 | 50.9 | **47.3** | 48.7 | 47.5 |
| I + D + W | 85.6 | 85.3 | **79.6** | 84.5 | 51.5 | **48.1** | 49.3 | **48.1** |
| I + L + R | 93.4 | **88.4** | 93.0 | 89.7 | 59.1 | 57.2 | 57.1 | 58.3 |
| I + L + W | 96.6 | **93.0** | 98.1 | **93.0** | 61.5 | **59.7** | 60.1 | 60.2 |
| I + R + W | 93.3 | **88.5** | 91.8 | 90.0 | 58.8 | 57.1 | **57.0** | 58.4 |
| D + L + R | 91.5 | 96.0 | **84.8** | 95.8 | 50.5 | 47.3 | 47.9 | 47.4 |
| D + L + W | 93.0 | 102.0 | **88.9** | 98.1 | 51.4 | 48.1 | 49.2 | **47.9** |
| D + R + W | 90.6 | 97.0 | **87.2** | 97.3 | 50.4 | 47.1 | 47.7 | 47.3 |
| L + R + W | 122.8 | 107.4 | 116.0 | **106.4** | 70.4 | 63.1 | 63.7 | **61.3** |
| I + D + L + R | 84.1 | 83.5 | **77.0** | 83.2 | 51.2 | **47.6** | 48.3 | **47.6** |
| I + D + L + W | 84.7 | 86.0 | **78.1** | 84.2 | 51.8 | 48.2 | 48.9 | **48.0** |
| I + D + R + W | 83.4 | 84.0 | **77.9** | 83.9 | 50.8 | 47.6 | 48.2 | 47.7 |
| I + L + R + W | 93.9 | **88.6** | 92.7 | 89.5 | 59.1 | 57.1 | **56.8** | 58.4 |
| D + L + R + W | 89.4 | 97.6 | **84.4** | 95.8 | 50.7 | **47.4** | 47.7 | **47.4** |
| I + D + L + R + W | 82.9 | 83.7 | **76.6** | 83.1 | 51.3 | **47.6** | 47.9 | **47.6** |

Table 5: Full results of the comparisons among Visual-Odometry Transformer based fusion , X-Fusion, and X-Fusion's variants.

## A.1 FEATURE EXTRACTOR PRE-TRAINING

As mentioned in the main work, to effectively extract informative high-dimensional representations from modal data, we designed modality-specific feature extractors whose backbone structures could be selected based on the data structure of each modality and the characteristics of the datasets. Meanwhile, we employ pre-training techniques to preserve the efficiency of feature extraction. Initially, for each modality, we train an entire model on the respective dataset and task. A portion of the model architecture is then repurposed as the feature extractor, with the corresponding pre-trained weights frozen during subsequent training. For instance, a ResNet-18 model was first trained on RGB images from the MM-Fi dataset to perform HPE. Subsequently, the MLP structure of the model's output layer was discarded, retaining only the bottleneck blocks serving as the feature extractor. To identify efficient backbone structures, we conducted experiments to validate the performance of various backbone structures on each dataset. The finalized backbones are presented below. When utilizing MM-Fi for HPE, we employ the ResNet-18 architecture (He et al., 2016) for both the RGB and Depth modalities, the Point Transformer architecture (Zhao et al., 2021) for LiDAR, the Point Transformer architecture without transition down layers for mmWave point cloud data, and the MetaFi++ architecture (Zhou et al., 2023) for Wifi-CSI. For HAR using MM-Fi, the same backbone architectures are utilized, with the parameters specifically trained for HAR due to the differing feature representation requirements across tasks. When utilizing the XRF55 dataset for HAR, the ResNet-18 architecture is adopted for the RFID, WiFi, and mmWave modalities to maintain consistency with the baseline backbone networks employed in the study (Wang et al., 2024).

| Ablation Studies on MM-Fi HAR | Dropout Layer | | Positional Encoding | |
|---|---|---|---|---|
| | Accuracy | | Accuracy | |
| Methods | DPR = 0.3 | DPR = 0.0 | False | True |
| I | 26.1% | **27.4%** | 25.4% | **27.4%** |
| D | 46.6% | **47.3%** | 47.0% | **47.3%** |
| L | 12.8% | **32.9%** | 31.9% | **32.9%** |
| R | 80.9% | **86.0%** | 85.5% | **86.0%** |
| I + D | **46.3%** | 45.2% | **45.9%** | 45.2% |
| I + L | 27.1% | **29.8%** | **30.6%** | 29.8% |
| I + R | 69.2% | **72.4%** | **74.4%** | 72.4% |
| D + L | 46.2% | **48.4%** | **49.0%** | 48.4% |
| D + R | 74.7% | **80.2%** | 78.6% | **80.2%** |
| L + R | 79.3% | **86.7%** | **87.4%** | 86.7% |
| I + D + L | **46.1%** | **46.1%** | **47.6%** | 46.1% |
| I + D + R | 67.6% | **70.8%** | 69.9% | **70.8%** |
| I + L + R | 66.9% | **74.4%** | **75.6%** | 74.4% |
| D + L + R | 72.6% | **80.1%** | 78.3% | **80.1%** |
| I + D + L + R | 65.8% | **70.0%** | **70.7%** | 70.0% |

Table 6: Ablation studies on drop out layers and positional encoding layer.

## A.2 EVALUATION OF MODALITY-SPECIFIC FEATURES COMBINING METHODS

Our objective is to assess whether concatenation is the most effective method for combining modality-specific features into multimodal embedding. To investigate this, we conducted a controlled experiment on the MM-Fi HPE task. In this experiment, we compared the concatenation method with the feature addition approach, where the feature values from each modality are summed and then divided by the number of modalities. As demonstrated in the Table 11, the feature addition approach failed to preserve distinctive modality features in the multimodal embeddings, resulting in average performance deteriorations of 15.7 mm on MPJPE and 5.9 mm on PA-MPJPE.

## A.3 COMPARISON WITH SOTA MULTIMODAL FUSION APPROACHES

To demonstrate the the multimodal fusion effectiveness of our proposed X-Fi, we conduct comparisons with SOTA multimodal fusion approaches that are based on fixed modality combinations, e.g. ImmFusion (Chen et al., 2023) and Meta-Transformer(Zhang et al., 2023). The experiments are conducted on the MM-Fi dataset(Yang et al., 2024) for the HPE task. For ImmFusion, we first train the model using the modality combination from the original paper (I+D+R) and then adapt the method to include all modalities (I+D+R+L+W). For Meta-Transformer, we use the Meta-Transformer-B16 encoder as the backbone. Initially, we fine-tune the pretrained encoder weights (from the original repository) and train a regression head on all modalities (I+D+R+L+W). Next, we freeze the encoder weights and fine-tune the regression head on Rgb+Depth+LiDAR modalities, obtaining the Meta-Transformer-B16F(I+D+L) result. Finally, we fine-tune both the encoder and the regression head on the three modalities to produce the Meta-Transformer-B16T(I+D+L) result. The results shown in the Table 7 demonstrate X-Fi's superior performance.

## A.4 ABLATION STUDIES

To demonstrate the effectiveness of our proposed modality-invariant foundation model and the corresponding optimization process, we evaluate the impact of the dropout layer, positional encoding, Key-Value generators, the optimizer choice, the number of iterations on X-Fusion block, and the number of layers in cross-modal transformer. The results are illustrated in the Table 6, 8,& 11.

**Dropout Layer.** To evaluate the effect of dropout layers, we incorporated a dropout layer with a rate of 0.3 into all transformer structures within X-Fi and assessed its performance on the MMFi dataset for the HAR task. The experimental results indicated that the introduction of the dropout

| Modalities | Methods | MPJPE (mm) | PA-MPJPE (mm) |
|---|---|---|---|
| Modalities | Methods | MPJPE (mm) | PA-MPJPE (mm) |
| I + D + L | Meta-Transformer-B16F | 105.2 | 70.3 |
| | Meta-Transformer-B16T | 96.2 | 68.1 |
| | X-Fusion | **84.8** | **48.2** |
| I + D+ R | ImmFusion | 207.3 | 146.8 |
| | X-Fusion | **83.4** | **47.3** |
| I + D + L + R + W | ImmFusion | 107.9 | 63.7 |
| | Meta-Transformer-B16T | 91.4 | 64.9 |
| | X-Fusion | **83.7** | **47.6** |

Table 7: Comparison between X-Fi and SOTA multimodal fusion approaches, e.g. ImmFusion and Meta-Transformer.

layer led to a 4.6% decrease in average accuracy. Furthermore, the dropout layer significantly impaired the model's predictive performance, particularly when handling challenging modalities.

**Optimizer Choice.** To assess the impact of different optimizer choices on experimental results, we conducted a set of controlled experiments using the SGD optimizer with a learning rate scheduler on the MM-Fi HPE task. The results indicated that the AdamW optimizer facilitated better convergence to a more optimal local minimum, leading to an average performance improvement of 50.9 mm on MPJPE and 44.6 mm on PA-MPJPE.

**Positional Encoding.** To assess the impact of LiDAR positional encoding, we conducted an experiment by removing the positional encoding structure. The results demonstrated that positional encoding provided a slight improvement in the performance of X-Fi with single-modality input. However, when faced with multi-modal inputs, the model's performance fluctuated, exhibiting worse results with certain modality combinations while performing better with others.

**Key-Value Generators.** To assess the impact of Key-Value Generators, we conducted an experiment by removing the Key-Value Generators and directly use modality-specific features to inject information in the cross-attention module. The ablation study results shown in the Table 8 proves that utilizing key-value generators to obtain key-value pairs is more efficient than directly using initial modality features as k-v.

| Ablation Studies on XRF55 HAR | Number of Iterations (#) | | Key-Value Generators (with/without) | | Depth of cross-modal transformer (#) | |
|---|---|---|---|---|---|---|
| Methods | # = 2 | # = 4 | without | with | # = 4 | # = 1 |
| R | 80.6% | **83.9%** | 82.0% | **83.9%** | 82.8% | **83.9%** |
| W | 24.2% | **55.7%** | 26.8% | **55.7%** | 54.8% | **55.7%** |
| RF | 38.8% | **42.5%** | 40.0% | **42.5%** | 40.5% | **42.5%** |
| R + W | 84.1% | **88.2%** | 85.0% | **88.2%** | 87.1% | **88.2%** |
| R + RF | 84.1% | **86.5%** | 84.7% | **86.5%** | 84.2% | **86.5%** |
| W + RF | 48.0% | **58.1%** | 50.2% | **58.1%** | 54.6% | **58.1%** |
| R + W + RF | 86.3% | **89.8%** | 86.7% | **89.8%** | 87.2% | **89.8%** |

Table 8: Ablation studies on Key-Value generators, number of iterations on X-Fusion block, and number of layers in cross-modal transformer.

**Number of iterations on X-Fusion block.** To assess the impact of the number of iterations on X-Fusion block, we conducted an experiment by adjusting the number. The ablation study results shown in the Table 8 demonstrate that when the number of iterations is set to 2, the model finds difficulties in digging and preserving features from weak modalities. We need to set the number of

iterations on the X-Fusion block to at least 4 to repeatedly inject information from weaker modalities, enabling the model to more robustly fit all participating modalities during training.

**Number of layers in cross-modal transformer.** To assess the impact of the number of layers in cross-modal transformer, we have added the ablation of the number of the layers (depth) in cross-modal transformers on the XRF55 HAR task. The default depth in our experiments is 1 and the depth we choose in the comparison experiment is 4. The results shown in the Table 8 indicate that adding the number of the layers in the cross-modal transformer will not yield better performance.

### A.5 HPE QUANTITATIVE RESULT INVESTIGATION

In order to have a more detailed look at the HPE quantitative result and show the observation that RGB modality captures better leg movements and depth modality is better for the arm, we have zoomed in the areas and the picture after enlargement has been added in the Figure 5. To seek the reason for such observation, we explore the characteristics of the rgb data and the depth data in MM-Fi. We find that due to the arm's movement typically being faster than the legs, combined with the insufficient frame rate of the camera, the arms in the RGB samples are often blurry, while the legs appear very clear. This results in poor RGB-based estimation performance for the arms. In contrast, in depth images, the arm contours are very clear, but the legs almost blend into the background, making their contours indistinct, which leads to poor depth-based estimation performance for the legs. We also include two images samples from rgb and depth in the Figure 6.

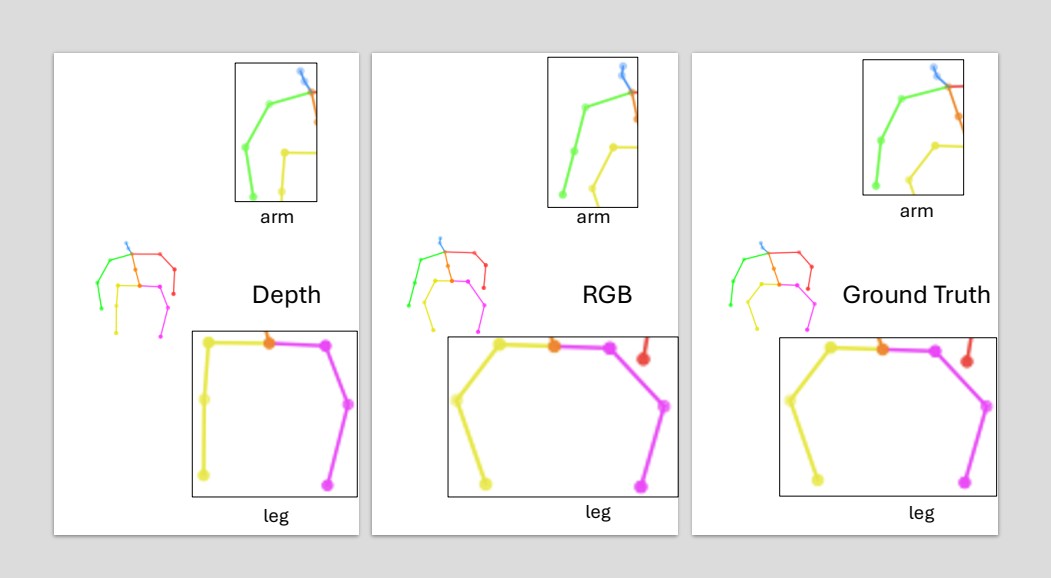

Figure 5: The detailed comparison among the rgb HPE result, the depth HPE result, and the ground truth.

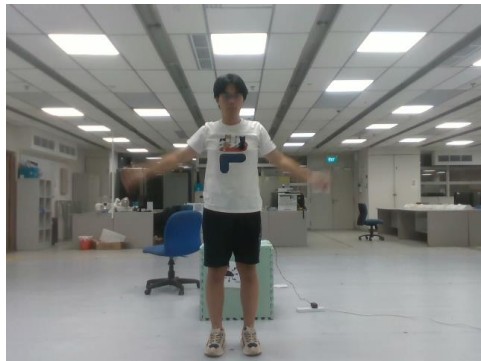 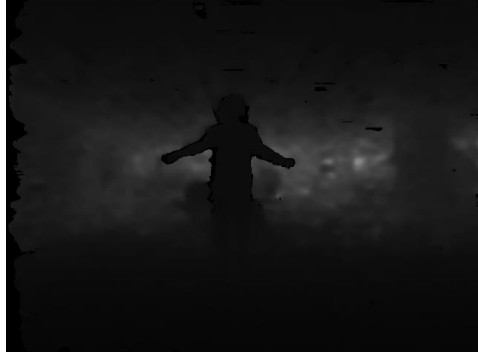

RGB Image Sample                     Depth Image Sample

Figure 6: The comparison between a rgb training sample and a depthtraining sample.

## A.6 CONTINUAL LEARNING ON X-FI

In order to validate the continual learning performance of X-Fi, we carry out experiment which gradually add new modality each time. We first trained an X-Fi model from scratch with RGB and Depth modalities. Then we added mmWave modality and fine-tuned the model based on the pretrained parameters. The results is shown in the Table 9.

| MM-Fi | MPJPE (mm) | | PA-MPJPE (mm) | |
|---|---|---|---|---|
| Methods | train with Rgb and Depth | Fine-tune with mmWave | train with Rgb and Depth | Fine-tune with mmWave |
| I | 117.9 | 105.2 | 71.7 | 64.3 |
| D | 108.5 | 130.2 | 56.2 | 52.8 |
| R | - | 128.1 | - | 72.4 |
| I + D | 99.2 | 106.4 | 59.2 | 53.6 |
| I + R | - | 92.8 | - | 58.7 |
| D + R | - | 115.7 | - | 50.3 |
| I + D + R | - | 100.8 | - | 51.8 |

Table 9: Continual Learning performance of X-Fi.

| Model Structure | Number of Parameters |
|---|---|
| X-Fusion block (5 modalities) | 17,903,667 |
| X-Fusion block (4 modalities) | 14,748,723 |
| X-Fusion block (3 modalities) | 11,593,779 |
| X-Fusion block (2 modalities) | 8,438,835 |
| X-Fusion block (1 modality) | 5,283,891 |
| VO Transformer fusion block | 12,636,723 |

Table 10: Comparison of number of parameters used in VO Transformer fusion block and X-Fusion block.

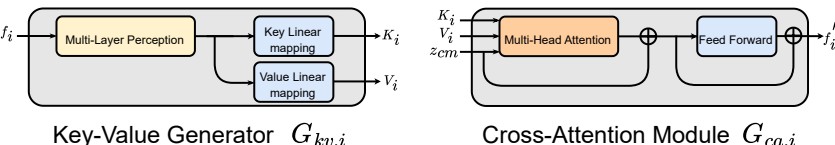

Key-Value Generator $G_{kv,i}$        Cross-Attention Module $G_{ca,i}$

Figure 7: The detailed architecture of the proposed key-value generators and cross-attention modules.

| Ablation Studies on MM-Fi HPE | multi-modal feature combining methods | | | | optimizer choices | | | |
|---|---|---|---|---|---|---|---|---|
| | MPJPE | | PA-MPJPE | | MPJPE | | PA-MPJPE | |
| Methods | Add | Concate | Add | Concate | SGD | AdamW | SGD | AdamW |
| I | 123.4 | **93.9** | 78 | **60.3** | 239.1 | **93.9** | 165.2 | **60.3** |
| D | 112.5 | **101.8** | 54.4 | **48.4** | 211.9 | **101.8** | 146.3 | **48.4** |
| L | 226.4 | **167.1** | 104.2 | **103.2** | 243.9 | **167.1** | 130.4 | **103.2** |
| R | 144.3 | **127.4** | 70.3 | **69.8** | 248.2 | **127.4** | 148.1 | **69.8** |
| W | 227.7 | **225.6** | **104.8** | 105.3 | 296.5 | **225.6** | 138 | **105.3** |
| I + D | 96.1 | **86.1** | 53.4 | **48.1** | 166.4 | **86.1** | 116.1 | **48.1** |
| I + L | 115.9 | **93** | 76.7 | **59.7** | 151.2 | **93** | 105.1 | **59.7** |
| I + R | 104.3 | **88.8** | 63.1 | **57.3** | 149.4 | **88.8** | 110.9 | **57.3** |
| I + W | 118.7 | **93** | 75.8 | **59.5** | 144.4 | **93** | 102.8 | **59.5** |
| D + L | 108.9 | **102.5** | 53.5 | **48.4** | 140.1 | **102.5** | 95.8 | **48.4** |
| D + R | 106.6 | **98** | 52.8 | **47.3** | 147.8 | **98** | 106.3 | **47.3** |
| D + W | 110.6 | **101.8** | 53.5 | **48.1** | 136.6 | **101.8** | 99.1 | **48.1** |
| L + R | 135.8 | **109.8** | 67.5 | **63.4** | 166.6 | **109.8** | 108.3 | **63.4** |
| L + W | 210.5 | **159.5** | 102.9 | **102.7** | 184.3 | **159.5** | 113.5 | **102.7** |
| R + W | 131.7 | **117.2** | 67.4 | **62.7** | 192.5 | **117.2** | 117.5 | **62.7** |
| I + D + L | 94.4 | **84.8** | 53.1 | **48.2** | 127 | **84.8** | 87.6 | **48.2** |
| I + D + R | 93.4 | **83.4** | 52.4 | **47.3** | 136.2 | **83.4** | 92.9 | **47.3** |
| I + D + W | 96 | **85.3** | 53 | **48.1** | 124.5 | **85.3** | 88 | **48.1** |
| I + L + R | 101.4 | **88.4** | 62.5 | **57.2** | 121.5 | **88.4** | 85.5 | **57.2** |
| I + L + W | 114.5 | **93** | 75.4 | **59.7** | 121 | **93** | 87.1 | **59.7** |
| I + R + W | 102.7 | **88.5** | 62.2 | **57.1** | 127 | **88.5** | 90.2 | **57.1** |
| D + L + R | 105.2 | **96** | 52.4 | **47.3** | 125.8 | **96** | 88.6 | **47.3** |
| D + L + W | 108.5 | **102** | 53.2 | **48.1** | 128.1 | **102** | 91 | **48.1** |
| D + R + W | 105.8 | **97** | 52.4 | **47.1** | 127.9 | **97** | 91.4 | **47.1** |
| L + R + W | 128.1 | **107.4** | 66.9 | **63.1** | 164.9 | **107.4** | 104.5 | **63.1** |
| I + D + L + R | 92.2 | **83.5** | 52.2 | **47.6** | 107.4 | **83.5** | 76.6 | **47.6** |
| I + D + L + W | 94.4 | **86** | 53 | **48.2** | 106.2 | **86** | 77.2 | **48.2** |
| I + D + R + W | 93.5 | **84** | 52.2 | **47.6** | 115.9 | **84** | 79.3 | **47.6** |
| I + L + R + W | 100.3 | **88.6** | 62.3 | **57.1** | 113.6 | **88.6** | 80.5 | **57.1** |
| D + L + R + W | 104.8 | **97.6** | 52.2 | **47.4** | 125 | **97.6** | 88.5 | **47.4** |
| I + D + L + R + W | 92.3 | **83.7** | 52.1 | **47.6** | 101.2 | **83.7** | 73.6 | **47.6** |

Table 11: Ablation studies on comparing multi-modal features combininng methods and optimizer choices.

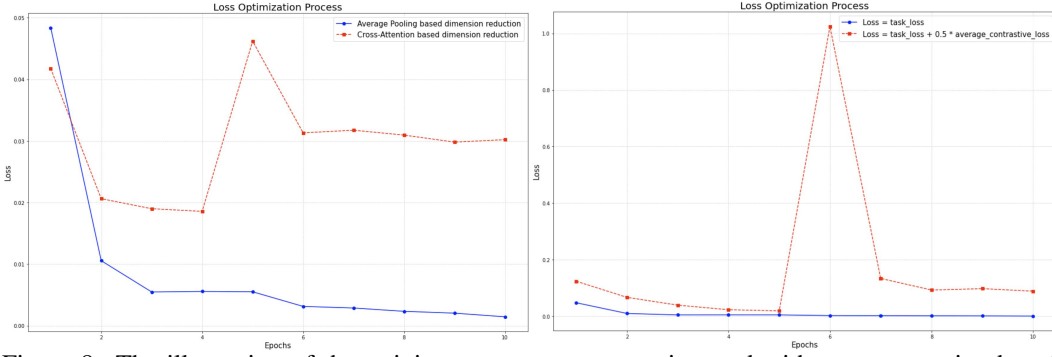

Figure 8: The illustration of the training processes we experimented with: cross-attention-based dimension reduction (left image) and the addition of contrastive loss (right image).

