# OpenReview forum: "X-Fi: A Modality-Invariant Foundation Model for Multimodal Human Sensing"
_ICLR.cc/2025/Conference — ICLR 2025 Poster_

### Official Review · Reviewer_rbmV · 2024-10-30

**Soundness:** 2
**Presentation:** 2
**Contribution:** 2
**Rating:** 6
**Confidence:** 3

**Summary:**

This paper introduces a multi-modal foundation model that integrates six distinct modalities to enhance human sensing capabilities. The proposed model demonstrates the scalability of various modality combinations and incorporates an X-Fusion block to fuse multi-modal features while preserving modality-specific information. The authors conduct extensive experiments on two multi-modal datasets to evaluate the model's performance in human pose estimation and human action recognition tasks.

**Strengths:**

1. The foundation model of different human sensing modalities is a promising approach, and the author made a meaningful attempt. The training schedule of this model is interesting

**Weaknesses:**

1. The section of problem definition is missing. This paper addresses two different tasks, which generally require different input formats of modalities. For instance, HAR requires video clips, but HPE could be video clips or frames. The clear details about the inputs and outputs are crucial for reader to understand.
2. There are some undefined notations, such as  in line 172 and  in Eq. 4.
3. The motivation regarding the module in subsection 3.2.2, which is to pay equal attention to each modality, is unconvincing. Different modalities offer varying degrees of human sensing information, and only valuable modality or information are important. The model paying more attention to dominant modalities is more intuitive. Equal attention could lead to the inclusion of irrelevant information and the potential omission of key details.
4. The cross-attention module is intended to accentuate modality-specific information. Nevertheless, the absence of ablation studies on this module precludes a conclusive evaluation of its efficacy, and there is no empirical evidence to substantiate the modality-specific features extracted into the K-V pairs. Some additional constraints are necessary. For instance, SimMMDG [1] leverages distance loss between modality-specific and modality-shared features and the orthogonal constraints is adopted in [2].
5. No ablation study on the components of proposed method.
6. The experimental results of Lidar in MM-Fi reveal inferior performance relative to mmWave in Tab. 1. This is in contrast to the results in MM-Fi under same experimental setup and backbones.  And it is also not intuitive as Lidar data, characterized by its higher point density, offers a more precise and detailed representation of the human compared to mmWave.
7. To ensure a fair comparison in Tab. 4, the number of parameters in the transformer and X-Fusion block should be equalized.

8.You employ multiple layers of cross-modal transformers to obtain cross-modal embeddings. How does the number of these iterative layers，N, impact the experimental results? Additionally, how might varying the number of transformer layers for different modality counts influence the outcomes?

9. The proposed method in the paper enforces a balance across all modalities. Could this forced balancing lead to a negative impact in tasks that primarily rely on a specific modality, especially if some modalities contain less informative data?

10. Performing computations across all modalities in each iteration could result in high computational complexity, particularly in multi-modal processing and long-sequence modeling. This increased demand could pose challenges for real-time applications, making practical deployment of the method more difficult.


[1]Dong H, Nejjar I, Sun H, et al. SimMMDG: A simple and effective framework for multi-modal domain generalization[J]. Advances in Neural Information Processing Systems, 2023, 36: 78674-78695.
[2]Jiang Q, Chen C, Zhao H, et al. Understanding and constructing latent modality structures in multi-modal representation learning[C]//Proceedings of the IEEE/CVF Conference on Computer Vision and Pattern Recognition. 2023: 7661-7671.

**Questions:**

Please see the weakness part.

---

> ### Author Response · Authors · 2024-11-21
> **Response to Reviewer rbmV (1/4)**
>
> We sincerely appreciate the reviewer rbmV for the detailed review, insightful questions and constructive comments. We are glad that the reviewer acknowledges “the meaningful attempt” we made and “**the promising approach**” we proposed. Here we answer all the questions and add extensive experiments for clarification. We hope the responses can address the concerns. In addition, we have submitted a revised manuscript with an appendix where we mark all the suggested tables, figures, and analytics in magenta color.
>
> **Q1: The section of problem definition is missing and the clear details about the inputs and outputs are crucial for reader to understand**
>
> **Answer**: We appreciate the suggestion and incorporated the problem definition of HPE and HAR tasks in section 3 line 151- 158, along with a detailed description of input formats in Section 4.1.
>
> The definition is:
> > Assume that input $X$ comprises a random combination of modalities within a specific range, including any number and type of modalities. Our objective is to learn a unified representation $z_{cm}$ that integrates modality-specific features from $X$ by using a modality-invariant foundation model $\mathcal{F}$. This representation can be mapped to various down-stream human sensing task spaces $t_{i}$ by using corresponding task-specific functions $g_{t_{i}}$, such that the task-specified loss function $L_{t_{i}}(g_{t_{i}}(\mathcal{F}(X)); y_{t_{i}})$ is minimized. In this paper, the $y_{t_{i}}$ refers to 3D human joints for HPE task and human activity classes for HAR task. Thus we propose X-Fi, a modality-invariant human sensing framework designed to map the arbitrary multimodal input $X$ to any task-specific target domain $t_{i}$.
>
> For each modality, the input format  remains consistent across all tasks. For instance, image frames are used in both HPE and HAR.
>
> **Q2: There are some undefined notations, such as in line 172 and in Eq. 4**
>
> **Answer**: We appreciate the corrections. We revised and unified all the notations in both the figures and the text throughout the paper.
>
> **Q3: Why encourage paying equal attention across modalities in subsection 3.2.2 when dominant modalities provide more valuable human sensing information and equal weighting could lead to the inclusion of irrelevant information?**
>
> **Answer**: We appreciate the suggestion. The motivation of our cross-attention module is that, previous studies [1] (CVPR 2022) & [2] (CVPR 2023) have shown if one modality disproportionately influences the model, the system will fail to generalize when certain modalities are distorted or missing.
>
> In [1], they reported the observation that ‘the Transformer models degrade dramatically with modal-incomplete data’. They trained a ViT on ‘full’ sets of image and text modalities and tested only with unimodal input. The result on MM-IMDb dataset reveals that the performance drops by 14.3% when missing image, and drops 36.7% when missing text.
>
> Similarly in [2], the trained modality-invariant transformer model marginally depends on the presence of depth modality. ‘Dropping RGB barely decreases performance, while dropping Depth causes the localization to fail more drastically’, as they described.
>
> Thus, balancing contributions allows the model to fully leverage complementary information across modalities and is essential to ensure robust and equitable performance across different scenarios.

---

> ### Author Response · Authors · 2024-11-21
> **Response to Reviewer rbmV (2/4)**
>
> **Q4: Are there any empirical evidence to substantiate the modality-specific features extracted into the K-V pairs. Some additional constraints are necessary. For instance, SimMMDG [3] leverages distance loss between modality-specific and modality-shared features.**
>
> **Answer**: We believe this suggestion is quite interesting so we added an experiment. In order to make sure that the modality-specific features are extracted into the K-V pairs and to align all modality-specific features and cross-modal embedding in a shared embedding space, we followed the method proposed in SimMMDG [3] to compute the average distance loss between each modality-specific features and cross-modal embedding. This distance loss was optimized jointly with the task loss during each training iteration.
>
> However, we found the model became hard to converge during training and it is easy to have gradient exploration issues. The training process is illustrated in appendix Figure 8.
>
> To tackle this issue, we tried to adjust the weighting parameter of the distance loss in the joint loss and found that the training became a bit more stable. However, there still exists a sudden gradient exploration issue after 13 training iterations.
>
> In conclusion, we realize that additional constraints between modality-specific features and cross-modal embedding is necessary to ensure effective modality-specific features extraction into the K-V pairs and could enhance model performance. However, further experiments and extensive hyperparameter tuning are required to determine how to introduce such constraints without compromising the stability of modality-invariant training.
>
> **Q5: Are there any ablation studies on the components of the proposed method, including the ablation study of the cross-attention module?**
>
> **Answer**: We would first apologize for not clearly explaining the transformer-based fusion comparison in section 4.2 and revised the section for better clarity. In fact, the transformer-based fusion we conducted could serve as an ablation study to evaluate the contribution of the cross-attention module. The transformer-based fusion is a reproduction of the SOTA modality-invariant method, VO Transformer [2], which removes the cross-attention module used for feature injection and only remains a cross-modal Transformer for feature fusion. The comparison results shown in Table 4 and Table 5 demonstrate the essential and effectiveness of the cross-attention module.  We have added the references here for better clarification.
>
> | MM-Fi HPE | MPJPE (mm) |  | PA-MPJPE (mm) |  |
> |---|---|---|---|---|
> | Method | VO Transformer [3] | X-Fi | VO Transformer [3] | X-Fi |
> | I | 99.6 | **93.9** | 65.0 | **60.3** |
> | L | 212.2 | **167.1** | 109.6 | **103.2** |
> | R | 138.0 | **127.4** | 74.6 | **69.8** |
> | W | 243.3 | **225.6** | 112.9 | **105.3** |
> | L + R | 123.4 | **109.8** | 69.9 | **63.4** |
> | L + W | 216.3 | **159.5** | 112.7 | **102.7** |
> | I + L + W | 96.6 | **93.0** | 61.5 | **59.7** |
>
> We also conducted further ablation studies on the key-value generators, the number of iterations on a X-Fusion iterative block, and the number of blocks of the X-Fusion stacked blocks. The results shown below are added to the appendix in the paper.
>
> The ablation study of key-value generators proves that utilizing key-value generators to obtain key-value pairs is more efficient than directly using initial modality features as k-v.
>
> | XRF55 HAR | Without KV Generators | With KV Generators |
> |---|---|---|
> | R | 82.0% | **83.9%** |
> | W | 26.8% | **55.7%** |
> | RF | 40.0% | **42.5%** |
> | R+W | 85.0% | **88.2%** |
> | R+RF | 84.7% | **86.5%** |
> | W+RF | 50.2% | **58.1%** |
> | R+W+RF | 86.7% | **89.8%** |
>
> The ablation study of the number of iterations on a X-Fusion iterative block is shown below.
>
> | XRF55 HAR | number of iteration = 2 | number of iteration = 4 |
> |---|---|---|
> | R | 80.6% | **83.9%** |
> | W | 24.2% | **55.7%** |
> | RF | 38.8% | **42.5%** |
> | R+W | 84.1% | **88.2%** |
> | R+RF | 84.1% | **86.5%** |
> | W+RF | 48.0% | **58.1%** |
> | R+W+RF | 86.3% | **89.8%** |
>
> We find that the performance with 4 iterations is better than the performance with 2 iterations across all modality combinations. The results indicate that when the number of iterations is set to 2, the model finds difficulties in digging and preserving features from weak modalities. We need to set the number of iterations on the X-Fusion block to at least 4 to repeatedly inject information from weaker modalities, enabling the model to more robustly fit all participating modalities during training.

---

> ### Author Response · Authors · 2024-11-21
> **Response to Reviewer rbmV (3/4)**
>
> **Q5: Are there any ablation studies on the components of the proposed method, including the ablation study of the cross-attention module?**
>
> **Answer**: The ablation study of  the number of blocks on the X-Fusion stacked block is shown below.
>
> | # of stacked X-Fusion blocks | number = 1 | number = 2 | number = 4 |
> |---|---|---|---|
> | I | 29.56% | 41.10% | **41.68%** |
> | D | 53.85% | **55.40%** | 52.65% |
> | R | **77.33%** | 73.61% | 73.16% |
> | I + D | 45.16% | 54.51% | **54.52%** |
> | I + R | 61.41% | **70.25%** | 67.36% |
> | D + R | **76.23%** | 75.28% | 73.89% |
> | I + D + R | 65.11% | 70.04% | **70.77%** |
>
> We find that one X-Fusion underfits on rgb modality and the results on rgb data is much worse than the other two variants' results. The results indicate that one X-Fusion block is not enough to effectively capture modality-specific features of each modality and at least two stacked X-Fusion blocks are required to achieve robust performance.
>
> However, we find no significant difference in performance between using two stacked blocks and four stacked blocks. Thus, increasing the number of stacked blocks further does not enhance performance, as the model converges to a stable optimal minimum.
>
> **Q6: Why is the X-Fi LiDAR single input result not as good as the results in MM-Fi, where Lidar achieves good performance?**
>
> **Answer**: We appreciate the question and conducted investigations on the problem. The reason is that we are carrying out modality invariant training on multiple modalities. Different modalities vary in their fitting complexity, and some may remain underfitted with the same number of training samples.
>
> For example, modalities like WiFi and LiDAR often require additional training samples to achieve an adequately fitted state. We could add more LiDAR data samples in the training process by adjusting the hyperparameters of the modality existence list to refine the model performance. For example, when we experimented on MM-Fi HAR task, we first set all modality existence probability to 80% and the model performed very poorly when facing a single LiDAR input. When we increased the LiDAR probability to 90% and reduced other modalities probabilities to 50%, the model performance improved a lot. After that, we further fine-tune the model by incorporating more LiDAR samples, and the single LiDAR result continues to improve. The comparison is illustrated in the table below.
>
> | MM-Fi HAR | ($P_{I}$,$P_{D}$,$P_{L}$,$P_{R}$) =  (0.8, 0.8, 0.8, 0.8) | ($P_{I}$,$P_{D}$,$P_{L}$,$P_{R}$) = (0.5,0.5,0.9,0.5) | Further Fine-Tuning |
> |---|---|---|---|
> | I | 26.3% | 27.4% | 26.8% |
> | D | 45.3% | 47.3% | 47.3% |
> | L | **3.8%** | **32.9%** | **52.6%** |
> | R | 79.2% | 86.0% | 84.5% |
>
> The performance of X-Fi on single LiDAR input continuously improves alongside adding more LiDAR training samples, as highlighted in the table above. This proves that we could adjust the hyperparameters of the modality existence list to add more specified training samples and refine the model performance.
>
> **Q7: To ensure a fair comparison in Tab. 4, the number of parameters in the transformer and X-Fusion block should be equalized.**
>
> **Answer**: We appreciate the suggestion. For fair comparison, we added the number of parameters for both the VO Transformer block and the X-Fusion block below.
>
> | Model Structure | Number of Parameters |
> |---|---|
> | X-Fusion block (5 modalities) | 17.9 M |
> | X-Fusion block (4 modalities) | 14.7 M |
> | X-Fusion block (3 modalities) | 11.6 M |
> | X-Fusion block (2 modalities) | 8.4 M |
> | X-Fusion block (1 modality) | 5.3 M |
> | VO Transformer fusion block | 12.6 M |
>
> We would emphasize that the number of active parameters for the VO Transformer block is fixed regardless of the number of modalities in the input combination. However, X-Fi has modality-specific structures that will be only activated when the corresponding modality is in the input and thus the number of parameters will be different for different numbers of modalities in the input combination. From the comparison, we can conclude that the superior performance achieved by X-Fusion is not achieved by only introducing more parameters.

---

> ### Author Response · Authors · 2024-11-21
> **Response to Reviewer rbmV (4/4)**
>
> **Q8: How does the number of the layers in cross-modal transformers impact the experimental results?**
>
> **Answer**: We appreciate the suggestion and added the ablation of the number of the layers (depth) in cross-modal transformers on the XRF55 HAR task. The default depth in our experiments is 1 and the depth we choose in the comparison experiment is 4. The results shown below indicate that adding the number of the layers in the cross-modal transformer will not yield better performance.
>
> | XRF55 | Depth = 4 | Depth = 1 |
> |---|---|---|
> | R | 82.8% | **83.9%** |
> | W | 54.8% | **55.7%** |
> | RF | 40.5% | **42.5%** |
> | R+W | 87.1% | **88.2%** |
> | R+RF | 84.2% | **86.5%** |
> | W+RF | 54.6% | **58.1%** |
> | R+W+RF | 87.2% | **89.8%** |
>
> **Q9: Could the equal attention across all modalities lead to a negative impact in tasks that primarily rely on a specific modality, especially if some modalities contain less informative data?**
>
> **Answer**: The cross-attention module we design would not lead to negative impact in certain tasks, which could be proven in the Table 4 & 5 that compare the X-Fusion results with Transformer-only results across all modality combinations.
>
> | MM-Fi HPE | MPJPE (mm) |  | PA-MPJPE (mm) |  |
> |---|---|---|---|---|
> | Methods | Tranfomer Only | X-Fusion | Tranfomer Only | X-Fusion |
> | I + W | 96.0 | **93.0** | 61.9 | **59.5** |
> | I + L + R | 93.4 | **88.4** | 59.1 | **57.2** |
> | I + L + R + W | 93.9 | **88.6** | 59.1 | **57.1** |
> | I + D + L + R + W | **82.9** | 83.7 | 51.3 | **47.6** |
>
> The goal of our method is not to enforce a rigid balance across all modalities but to mitigate the risk of over-reliance on dominant modalities, which could result in the loss of valuable information from weaker modalities.
>
> Our cross-attention fusion module achieves this by reintegrating overlooked modality-specific information into the cross-modal embeddings, thereby encouraging the model to also learn how to extract features from weak modalities during the modality invariant training process.
>
> This process does not diminish the model's ability to prioritize and extract features from more informative modalities, which is achieved by the cross-modal transformer structure. Instead, it encourages the model to learn richer and more complementary cross-modal features, ultimately improving its performance across diverse scenarios. This balanced approach ensures robustness and adaptability without compromising the model's effectiveness in tasks that heavily rely on specific modalities.
>
> **Q10: Computing all modalities in each iteration increases complexity, especially in multi-modal and long-sequence modeling, posing challenges for real-time applications and practical deployment.**
>
> **Answer**: Actually our inference complexity is only related to the number of modalities we use in a given scenario. For example, the X-Fi is trained on 5 modalities, but in the inference only the radar is used. Then only the radar encoder, cross-attention module, and key-value generator are activated so the complexity is still acceptable.
>
> The training complexity is not increased with the number of modality increased in each iteration, because for each iteration, we train X-Fi on a specific number of modalities and discard other modalities. This mimics the situation when some modalities are missing, enhancing the robustness of X-Fi and controlling the complexity for each iteration. Once the X-Fi model is trained, the model will only compute the corresponding parts for the modalities utilized in the specific application scenario. However, we admit that more iterations might be demanded for the convergence of X-Fi when more modalities are considered.
>
> **Reference**:
>
> [1] Ma, Mengmeng, Jian Ren, Long Zhao, Davide Testuggine, and Xi Peng. "Are multimodal transformers robust to missing modality?." In Proceedings of the IEEE/CVF Conference on Computer Vision and Pattern Recognition, pp. 18177-18186. 2022.
>
>
> [2] Memmel, Marius, Roman Bachmann, and Amir Zamir. "Modality-invariant Visual Odometry for Embodied Vision." In Proceedings of the IEEE/CVF Conference on Computer Vision and Pattern Recognition, pp. 21549-21559. 2023.
>
>
> [3] Dong, Hao, Ismail Nejjar, Han Sun, Eleni Chatzi, and Olga Fink. "SimMMDG: A simple and effective framework for multi-modal domain generalization." Advances in Neural Information Processing Systems 36 (2023): 78674-78695.

---

> ### Author Response · Authors · 2024-11-24
> **Looking forward to your reply**
>
> Dear reviewer rbmV,
>
> As the rebuttal deadline is approaching in **2 days**, could you please have a look at our response? We are looking forward to your reply!
>
> Feel free to let us know if you have any other concerns. Thanks for your time and effort!
>
> Best Regards,
>
> Authors of Submission 6113

---

> ### Author Response · Authors · 2024-11-26
> **Looking forward to your reply**
>
> Dear reviewer rbmV,
>
> As the rebuttal deadline is approaching **within a day**, could you please have a look at our response? We are looking forward to your reply!
>
> Feel free to let us know if you have any other concerns. Thanks for your time and effort!
>
> Best Regards,
>
> Authors of Submission 6113

---

> > ### Comment · Reviewer_rbmV · 2024-11-26
> > **Rebuttal**
> >
> > Thanks for the efforts in the response. The rebuttal address most of my concerns and I am willing to raise my final rating.

---

> > > ### Author Response · Authors · 2024-11-26
> > > **Appreciation for the constructive comments**
> > >
> > > We sincerely appreciate your helpful suggestions! Thanks for your time and effort!

---

### Official Review · Reviewer_u1Rk · 2024-11-01

**Soundness:** 3
**Presentation:** 3
**Contribution:** 3
**Rating:** 6
**Confidence:** 5

**Summary:**

This paper introduces a modality-invariant foundation model that allows for the integration of various sensor modalities for human sensing. The proposed X-Fi model employs a transformer architecture with a "X-fusion" mechanism to preserve modality-specific features in the multimodal integration process. The proposed model is designed to support any combination of sensor modalities with flexibility of the input dimension without requiring additional retraining. They evaluated the proposed method on two benchmark datasets, MM-Fi and XRF55, covering six distinct modalities, with two downstream tasks: human pose estimation and human activity recognition. The experimental results showed that the proposed X-Fi model improved the performance of HPE and HAR.

**Strengths:**

+ This paper addresses the important problem of modality-invariant human sensing by designing a foundational model with flexiblity without retraining.
+ The paper is clearly written and well-structured, effectively communicating the motivation, methodology, and results. Concepts such as modality-specific feature extraction, cross-attention fusion, and the role of the X-fusion block are clearly explained, The transformer-based architecture handles multimodal input data and X-fusion block is used for cross-modal feature fusion by using cross-attention.

**Weaknesses:**

- The paper lacks a thorough investigation of related cross-modal / modality-invariant foundational models, such as CLIP [1], PaLM-E [2], Meta-Transformer [3], ImageBind [4], IMU2CLIP [5], and zero-shot learning for HAR [6]. The authors need to investigate recent cross-modal foundational models.
- While the paper proposes the X-fusion block for multimodal fusion, the core approach of using transformers and cross-attention for feature fusion is not new. Similar attention-based fusion methods are widely used in the literature for feature-level integration. The paper could strengthen the novel contributions and clarify the novel aspects of X-fusion compared to existing methods.
- The evaluation experiments focus on comparing the proposed method with basic feature and decision-level fusion approaches. The proposed foundational model may be compared with other cross-modal foundational models.
- The evaluation was conducted on MM-Fi and XRF55 datasets, which primarily include specific modalities like cameras, WiFi, LiDAR. However, human activity recognition often involves wearable sensors like IMUs. It is better to add one or two more benchmark datasets with IMU sensors.


[1] Radford, Alec, Jong Wook Kim, Chris Hallacy, Aditya Ramesh, Gabriel Goh, Sandhini Agarwal, Girish Sastry et al. "Learning transferable visual models from natural language supervision." In International conference on machine learning, pp. 8748-8763. PMLR, 2021.
[2] Driess, Danny, Fei Xia, Mehdi SM Sajjadi, Corey Lynch, Aakanksha Chowdhery, Brian Ichter, Ayzaan Wahid et al. "PaLM-E: An Embodied Multimodal Language Model." In International Conference on Machine Learning, pp. 8469-8488. PMLR, 2023.
[3] Zhang, Yiyuan, Kaixiong Gong, Kaipeng Zhang, Hongsheng Li, Yu Qiao, Wanli Ouyang, and Xiangyu Yue. "Meta-transformer: A unified framework for multimodal learning." arXiv preprint arXiv:2307.10802 (2023).
[4] Girdhar, Rohit, Alaaeldin El-Nouby, Zhuang Liu, Mannat Singh, Kalyan Vasudev Alwala, Armand Joulin, and Ishan Misra. "Imagebind: One embedding space to bind them all." In Proceedings of the IEEE/CVF Conference on Computer Vision and Pattern Recognition, pp. 15180-15190. 2023.
[5] Moon, Seungwhan, Andrea Madotto, Zhaojiang Lin, Aparajita Saraf, Amy Bearman, and Babak Damavandi. "IMU2CLIP: Language-grounded Motion Sensor Translation with Multimodal Contrastive Learning." In Findings of the Association for Computational Linguistics: EMNLP 2023, pp. 13246-13253. 2023.
[6] Tong, Catherine, Jinchen Ge, and Nicholas D. Lane. "Zero-shot learning for imu-based activity recognition using video embeddings." Proceedings of the ACM on Interactive, Mobile, Wearable and Ubiquitous Technologies 5, no. 4 (2021): 1-23.

**Questions:**

1. Can the authors provide additional details on the distinguished contribution points of the X-fusion block compared to existing cross-attention fusion approaches?
2. Would the authors consider conducting further experiments with other cross-modal foundational models (e.g., Meta-Transformer, ImageBind) as baselines to provide a more comprehensive comparison?
3. Is it possible to include additional benchmark datasets with IMU data for HAR task?

---

> ### Author Response · Authors · 2024-11-21
> **Response to Reviewer u1Rk (1/2)**
>
> We sincerely appreciate the reviewer u1Rk for the insightful questions and constructive comments. We are glad that the reviewer appreciates that **“the problem we address is important”** and **“the paper is clearly written and well-structured”**. We address the lack of comparisons to prior methods and add experiments on other SOTA methods to prove the effectiveness of our proposed X-Fi. We also address the lack of evaluating X-Fi on datasets containing IMU sensor and add experiments on the mRI dataset [6] to prove the proposed X-Fi could be expanded to wearable sensors. In addition, we have submitted a revised manuscript with an appendix where we mark all the suggested tables, figures, and analytics in magenta color.
>
> **Q1: Add the following citation and briefly investigate the related cross-modal / modality-invariant foundational models.**
>
> **Answer**: We appreciate the suggestion and conduct literature reviews on the recommended papers. The discussion about these approaches and the corresponding citations are included in the related work section 2.2.
>
> **Q2: Can the authors provide additional details on the distinguished contribution points of the X-fusion block compared to existing cross-attention fusion approaches?**
>
> **Answer**: Actually cross-attention module design is based on existing cross-attention fusion works. It is a revamped module yet it does not denote our main novelty. The novelty of this paper is to address a practical problem in the human sensing research community: in the real-world sensing scenario, we always demand a single model that is trained only once and then used with different modality combinations of inputs for different applications, without the need for retraining multiple models.
>
> Several gaps are filled by the proposed X-Fi framework, including handling arbitrary number of input modalities, capturing and preserving unique characteristics of each modality, retaining each modality’s distinct features during multimodal fusion, and designing a unified training strategy to enable modality invariant human sensing. Cross-attention module is incorporated specifically to retain each modality’s distinct features during multimodal fusion. Therefore, we slightly modified the UniFusion module in the work UniRef++ [1] and validated its effectiveness on two large human sensing datasets, demonstrating performance improvements.
>
> **Q3: Would the authors consider conducting further experiments with other cross-modal foundational models (e.g., Meta-Transformer, ImageBind) as baselines to provide a more comprehensive comparison?**
>
> **Answer**: We appreciate the suggestion. We added experiments with Meta-Transformer [4] to compare our proposed X-Fi with SOTA multimodal fusion approaches that are based on fixed modality combinations. The experiments were conducted on the MM-Fi dataset [5] for the HPE task.
>
> In experiments, we used the Meta-Transformer-B16 encoder as the backbone. Initially, we fine-tuned the pretrained encoder weights (from the original repository) and trained a regression head on all modalities (I+D+R+L+W). Next, we froze the encoder weights and fine-tuned the regression head on I+D+L modalities, obtaining the Meta-Transformer-B16F result. Finally, we fine-tuned both the encoder and the regression head on I+D+L modalities to produce the Meta-Transformer-B16T result.The results shown below demonstrate X-Fi’s superior performance compared to meta-transformer.
>
> | MM-Fi | MPJPE (mm) |  |  | PA-MPJPE (mm) |  |  |
> |---|---|---|---|---|---|---|
> | Method | Meta Transformer-B16F [4] | Meta Transformer-B16T [4] | X-Fusion | Meta Transformer-B16F [4] | Meta Transformer-B16T [4] | X-Fusion |
> | I + D + L | 105.2 | 96.2 | **84.8** | 70.3 | 68.1 | **48.2** |
> | I+D+L+R+W | - | 91.4 | **83.7** | - | 64.9 | **47.6** |
>
> Additionally, we have already compared our method with the VO Transformer [2] in the original manuscript Section 4.3. The transformer based fusion method is the VO Transformer structure, and we have added the references here for better clarification. The experiment results have been listed in Table 4 and Table 5 and show our proposed X-Fi perform better on all modality combinations.
>
> | MM-Fi HPE | MPJPE (mm) |  | PA-MPJPE (mm) |  |
> |---|---|---|---|---|
> | Method | VO Transformer [2] | X-Fi | VO Transformer [2] | X-Fi |
> | I | 99.6 | **93.9** | 65.0 | **60.3** |
> | L | 212.2 | **167.1** | 109.6 | **103.2** |
> | R | 138.0 | **127.4** | 74.6 | **69.8** |
> | W | 243.3 | **225.6** | 112.9 | **105.3** |
> | L + R | 123.4 | **109.8** | 69.9 | **63.4** |
> | L + W | 216.3 | **159.5** | 112.7 | **102.7** |
> | I + L + W | 96.6 | **93.0** | 61.5 | **59.7** |

---

> ### Author Response · Authors · 2024-11-21
> **Response to Reviewer u1Rk (2/2)**
>
> **Q3: Would the authors consider conducting further experiments with other cross-modal foundational models (e.g., Meta-Transformer, ImageBind) as baselines to provide a more comprehensive comparison?**
>
> **Answer**: To better demonstrate the significance of our proposed X-Fi. We added further experiments with ImmFusion [3]. We first trained the model using the modality combination from the original paper (I+D+R) and then adapted the method to include all modalities (I+D+R+L+W). The results are shown below and indicate the superior performance of our proposed X-Fi compared to ImmFusion.
>
> | MM-Fi | MPJPE (mm) |  | PA-MPJPE (mm) |  |
> |---|---|---|---|---|
> | Method | ImmFusion [3] | X-Fusion | ImmFusion [3] | X-Fusion |
> | I + D + R | 207.3 | **83.4** | 146.8 | **47.3** |
> | I + D + L + R + W | 107.9 | **83.7** | 63.7 | **47.6** |
>
> **Q4: Is it possible to include additional benchmark datasets with IMU data for HAR task?**
>
> **Answer**: Yes, we conducted initial experiments on mRI dataset [6] that includes IMU data.
>
> Our current motivation is centered on passive sensing from a single perspective in order for application scenarios such as autonomous driving and robotics sensing. Thus, all the modalities we utilized are based on passive sensing and the datasets we used are based on these sensors. But our method is easy to be expanded to active perception sensors by incorporating more modality-specific feature tokens into the modality invariant training process.
>
> In the experiment, we used protocol 1 and setting 1 utilized in the original paper and the results shown below demonstrate the superior performance of our proposed X-Fi compared with the baseline.
>
> | mRI [6] | MPJPE (mm) |  | PA-MPJPE (mm) |  |
> |---|---|---|---|---|
> |  | Baseline | X-Fi | Baseline | X-Fi |
> | IMUs | 80.2 | **77.5** | 51.9 | **47.3** |
> | IMUs+mmWave | 105.6 | **84.2** | 69.8 | **52.4** |
>
> **Reference:**
>
> [1] Wu, Jiannan, Yi Jiang, Bin Yan, Huchuan Lu, Zehuan Yuan, and Ping Luo. "UniRef++: Segment Every Reference Object in Spatial and Temporal Spaces." arXiv preprint arXiv:2312.15715 (2023).
>
> [2] Memmel, Marius, Roman Bachmann, and Amir Zamir. "Modality-invariant Visual Odometry for Embodied Vision." In Proceedings of the IEEE/CVF Conference on Computer Vision and Pattern Recognition, pp. 21549-21559. 2023.
>
> [3] Chen, Anjun, Xiangyu Wang, Kun Shi, Shaohao Zhu, Bin Fang, Yingfeng Chen, Jiming Chen, Yuchi Huo, and Qi Ye. "Immfusion: Robust mmwave-rgb fusion for 3d human body reconstruction in all weather conditions." In 2023 IEEE International Conference on Robotics and Automation (ICRA), pp. 2752-2758. IEEE, 2023.
>
> [4] Zhang, Yiyuan, Kaixiong Gong, Kaipeng Zhang, Hongsheng Li, Yu Qiao, Wanli Ouyang, and Xiangyu Yue. "Meta-transformer: A unified framework for multimodal learning." arXiv preprint arXiv:2307.10802 (2023).
>
> [5] Yang, Jianfei, He Huang, Yunjiao Zhou, Xinyan Chen, Yuecong Xu, Shenghai Yuan, Han Zou, Chris Xiaoxuan Lu, and Lihua Xie. "Mm-fi: Multi-modal non-intrusive 4d human dataset for versatile wireless sensing." Advances in Neural Information Processing Systems 36 (2024).
>
> [6] An, Sizhe, Yin Li, and Umit Ogras. "mri: Multi-modal 3d human pose estimation dataset using mmwave, rgb-d, and inertial sensors." Advances in Neural Information Processing Systems 35 (2022): 27414-27426.

---

> ### Author Response · Authors · 2024-11-23
> **Looking forward to your reply**
>
> Dear reviewer u1Rk,
>
> As the rebuttal deadline is approaching, could you please have a look at our response? We are looking forward to your reply!
>
> Feel free to let us know if you have any other concerns. Thanks for your time and effort!
>
> Best Regards,
>
> Authors of Submission 6113

---

> ### Author Response · Authors · 2024-11-24
> **Looking forward to your reply**
>
> Dear reviewer u1Rk,
>
> As the rebuttal deadline is approaching in **2 days**, could you please have a look at our response? We are looking forward to your reply!
>
> Feel free to let us know if you have any other concerns. Thanks for your time and effort!
>
> Best Regards,
>
> Authors of Submission 6113

---

> > ### Comment · Reviewer_u1Rk · 2024-11-25
> >
> > Thank you for the authors' response.
> > The authors address my comments and mostly answer my concerns.

---

> ### Author Response · Authors · 2024-11-25
> **Thanks and kind request for rate increment**
>
> Dear reviewer u1Rk,
>
> Thank you for acknowledging that our rebuttal has mostly addressed your initial concerns. We greatly appreciate your engagement and constructive comments.
>
> Currently three reviewers have responded that the rebuttal has addressed their concerns and may increase the score, (while two reviewers have not responded yet). It is believed that the revised manuscript has been **signficantly improved by address your concerns and adding many experimental results and analysis**. If so, we kindly request if you could consider to increase the rating of our submission, as the current score still remains below the borderline.
>
> Thank you so much for your time and consideration.
>
> Best regards,
>
> Authors of Submission 6113

---

> ### Author Response · Authors · 2024-11-27
> **Appreciation for the constructive comments**
>
> We sincerely appreciate your helpful suggestions! Thanks for your time and effort!

---

### Official Review · Reviewer_Rmx6 · 2024-11-01

**Soundness:** 2
**Presentation:** 3
**Contribution:** 3
**Rating:** 8
**Confidence:** 4

**Summary:**

In this paper, the authors propose a multi-modal modality invariant foundation model to tackle the problem in the current human sensing techniques. They introduced X-Fi block to combine the feature representation from different modalities.  The authors utilized two datasets to verify the effectiveness of the proposed model on human action recognition and pose estimation. The proposed model outperformed the other methods.

**Strengths:**

The paper presents a simple yet effective framework for multi-modal fusion from heterogenous modalities.
Two large datasets were adopted in the experiments.
The authors conducting experiments on  different modality combinations.

**Weaknesses:**

1- The paper needs polishing and revision. For example in the first page (line 043) "Alternative modalities like LiDAR, mmWave radar,
and WiFi have be introduced to overcome these challenges" --> it should been not be. Additionally, the authors are mixing Fig with Figure. Or capital letters within the sentence--> Line 233 "Notably, The multi-head" and Line 259.
2- The authors mentioned in the contribution and abstract the proposed model surpassed the SOTA. I did not find a table of comparison against the SOTA especially the only comparison is against baseline.
3- The authors did not provide the experimental setting. For example, MM-Fi [1] has three settings and it is not clear which setting is adopted. I recommend to adopt the three settings.


[1]Yang, Jianfei, He Huang, Yunjiao Zhou, Xinyan Chen, Yuecong Xu, Shenghai Yuan, Han Zou, Chris Xiaoxuan Lu, and Lihua Xie. "Mm-fi: Multi-modal non-intrusive 4d human dataset for versatile wireless sensing." Advances in Neural Information Processing Systems 36 (2024).

**Questions:**

1- I suggest to have a joint loss ( L= L_I+ L_R+L_W+....L_N). In other words, to have loss for each modality and the X-Fusion output as well. It should boost the performance.
2-   I suggest an ablation study on the number of  stacked blocks in X-Fi  block.
3- I was expecting an experiment which gradually add new modality each time. For example the model is trained first on RGB then WiFI is  added by fixing the parameters and finetuning the model after adding the new modality.

---

> ### Author Response · Authors · 2024-11-21
> **Response to Reviewer Rmx6 (1/2)**
>
> We sincerely appreciate the reviewer Rmx6 for the insightful questions and constructive comments. We are glad that the reviewer acknowledges **"the simple yet effective framework"** we proposed. We address the lack of comparisons to prior methods and add experiments on other SOTA methods to prove the effectiveness of our proposed X-Fi. We also add experiments as suggested to boost the performance of X-Fi. In addition, we have submitted a revised manuscript with an appendix where we mark all the suggested tables, figures, and analytics in magenta color.
>
> **Q1: The paper needs polishing and revision.**
>
> **Answer**: We appreciate the suggestions. We revised the paper carefully and made corrections to polish the paper.
>
> **Q2: Are there any comparisons against the SOTA method? Why only compare with baseline?**
>
> **Answer**: The two baselines we adopted are originally proposed in the MM-Fi [1] and the XRF55 [2] dataset papers. And these results are the SOTA results so far on these datasets after searching through all the multimodal fusion research works based on these datasets.
>
> For modality invariant SOTA method VO Transformer [1], we have already compared our method with VO Transformer in the original manuscript Section 4.3. The transformer based fusion method is the VO Transformer structure, and we have added the references here for better clarification. The experiment results have been listed in Table 4 and Table 5.
>
> | MM-Fi HPE | MPJPE (mm) |  | PA-MPJPE (mm) |  |
> |:---:|:---:|:---:|:---:|:---:|
> | Method | VO Transformer [3] | X-Fi | VO Transformer [3] | X-Fi |
> | I | 99.6 | **93.9** | 65.0 | **60.3** |
> | L | 212.2 | **167.1** | 109.6 | **103.2** |
> | R | 138.0 | **127.4** | 74.6 | **69.8** |
> | W | 243.3 | **225.6** | 112.9 | **105.3** |
> | L + R | 123.4 | **109.8** | 69.9 | **63.4** |
> | L + W | 216.3 | **159.5** | 112.7 | **102.7** |
> | I + L + W | 96.6 | **93.0** | 61.5 | **59.7** |
>
> As suggested, to better demonstrate the significance of our proposed X-Fi, we expand the ImmFusion method by concatenating multimodal features and expand the VO Transformer method by feeding all modalities’ features into a single transformer structure, which are then compared to our method on the benchmark.
>
> To better demonstrate the significance of our proposed X-Fi. We added experiments with ImmFusion [4] and Meta-Transformer [5] to compare our proposed X-Fi with SOTA multimodal fusion approaches that are based on fixed modality combinations. The experiments were conducted on the MM-Fi dataset [1] for the HPE task.
>
> For ImmFusion [4], we first trained the model using the modality combination from the original paper (I+D+R) and then adapted the method to include all modalities (I+D+R+L+W). The results are shown below and indicate the superior performance of our proposed X-Fi compared to ImmFusion.
>
> | MM-Fi | MPJPE (mm) |  | PA-MPJPE (mm) |  |
> |:---:|:---:|:---:|:---:|:---:|
> | Method | ImmFusion [4] | X-Fusion | ImmFusion [4] | X-Fusion |
> | I + D + R | 207.3 | **83.4** | 146.8 | **47.3** |
> | I + D + L + R + W | 107.9 | **83.7** | 63.7 | **47.6** |
>
> For Meta-Transformer [5], we used the Meta-Transformer-B16 encoder as the backbone to carry out experiments. The results shown below demonstrate X-Fi’s superior performance.
>
> | MM-Fi | MPJPE (mm) |  |  | PA-MPJPE (mm) |  |  |
> |:---:|:---:|:---:|:---:|:---:|:---:|:---:|
> | Method | Meta Transformer-B16F [5] | Meta Transformer-B16T [5] | X-Fusion | Meta Transformer-B16F [5] | Meta Transformer-B16T [5] | X-Fusion |
> | I + D + L | 105.2 | 96.2 | **84.8** | 70.3 | 68.1 | **48.2** |
> | I+D+L+R+W | - | 91.4 | **83.7** | - | 64.9 | **47.6** |
>
> **Q3: What is the dataset setting used in the experiments?**
>
> **Answer**: We described the dataset details and added the dataset split settings descriptions in section 4.1 line 353 - 354 . The split setting we used for MM-Fi is S1 Random Split and the original split setting for XRF55, as outlined in their respective papers.

---

> ### Author Response · Authors · 2024-11-21
> **Response to Reviewer Rmx6 (2/2)**
>
> **Q4: I suggest having a joint loss that includes loss for each modality and the X-Fusion output.**
>
> **Answer**: We believe this suggestion is quite interesting so we added an experiment. In order to align each modality-specific features and cross-modal embedding, we followed the method proposed in SimMMDG [3] to compute the average distance loss between each modality-specific features and cross-modal embedding. This distance loss was optimized jointly with the task loss during each training iteration.
>
> However, we found the model became hard to converge during training and it is easy to have gradient explosion issues. The training process is illustrated in appendix Figure 8.
>
> To tackle these issues, we tried to adjust the weighting parameter of the distance loss in the joint loss and found that the training became a bit more stable. However, there still exists a sudden gradient explosion issue after 13 training iterations.
>
> In conclusion, we realize that additional loss between modality-specific features and cross-modal embedding is necessary to align each modality features to a shared embedding space and could enhance model performance. However, further experiments and extensive hyperparameter tuning are required to determine how to introduce such alignment without compromising the stability of modality-invariant training.
>
> **Q5: I suggest an ablation study on the number of stacked blocks in X-Fi block.**
>
> **Answer**: We appreciate the suggestion and conducted an ablation study on the number of stacked blocks on MM-Fi HAR task.
>
> | # of stacked X-Fusion blocks | number = 1 | number = 2 | number = 4 |
> |---|---|---|---|
> | I | 29.56% | 41.10% | **41.68%** |
> | D | 53.85% | **55.40%** | 52.65% |
> | R | **77.33%** | 73.61% | 73.16% |
> | I + D | 45.16% | 54.51% | **54.52%** |
> | I + R | 61.41% | **70.25%** | 67.36% |
> | D + R | **76.23%** | 75.28% | 73.89% |
> | I + D + R | 65.11% | 70.04% | **70.77%** |
>
> We find that one X-Fusion underfits on rgb modality and the results on rgb data is much worse than the other two variants' results. The results indicate that one X-Fusion block is not enough to effectively capture modality-specific features of each modality and at least two stacked X-Fusion blocks are required to achieve robust performance.
>
> However, we find no significant difference in performance between using two stacked blocks and four stacked blocks. Thus, increasing the number of stacked blocks further does not enhance performance, as the model converges to a stable optimal minimum.
>
> **Q6: I am expecting an experiment which gradually adds new modality each time.**
>
> **Answer**: We believe this suggestion is quite interesting so we added an experiment. We first trained an X-Fi model from scratch with RGB and Depth modalities. Then we added mmWave modality and fine-tuned the model based on the pretrained parameters. The results are added in appendix Table 9 and shown below.
>
> |  | MPJPE (mm) |  | PA-MPJPE (mm) |  |
> |---|---|---|---|---|
> | MM-Fi | Train with Rgb and Depth | Finetune with mmWave | Train with Rgb and Depth | Finetune with mmWave |
> | I | 117.9 | 105.2 | 71.7 | 64.3 |
> | D | 108.5 | 130.2 | 56.2 | 52.8 |
> | R | - | 128.1 | - | 72.4 |
> | I + D | 99.2 | 106.4 | 59.2 | 53.6 |
> | I + R | - | 92.8 | - | 58.7 |
> | D + R | - | 115.7 | - | 50.3 |
> | I + D + R | - | 100.8 | - | 51.8 |
>
> The results show similar performance after adding mmWave modality and fine-tuning the pretrained model. This indicates that X-Fi could be easily expanded to newly introduced modalities without harming the performance on the existing modalities.
>
> **Reference:**
>
> [1] Yang, Jianfei, He Huang, Yunjiao Zhou, Xinyan Chen, Yuecong Xu, Shenghai Yuan, Han Zou, Chris Xiaoxuan Lu, and Lihua Xie. "Mm-fi: Multi-modal non-intrusive 4d human dataset for versatile wireless sensing." Advances in Neural Information Processing Systems 36 (2024).
>
> [2] Wang, Fei, Yizhe Lv, Mengdie Zhu, Han Ding, and Jinsong Han. "XRF55: A Radio Frequency Dataset for Human Indoor Action Analysis." Proceedings of the ACM on Interactive, Mobile, Wearable and Ubiquitous Technologies 8, no. 1 (2024): 1-34.
>
> [3] Memmel, Marius, Roman Bachmann, and Amir Zamir. "Modality-invariant Visual Odometry for Embodied Vision." In Proceedings of the IEEE/CVF Conference on Computer Vision and Pattern Recognition, pp. 21549-21559. 2023.
>
> [4] Chen, Anjun, Xiangyu Wang, Kun Shi, Shaohao Zhu, Bin Fang, Yingfeng Chen, Jiming Chen, Yuchi Huo, and Qi Ye. "Immfusion: Robust mmwave-rgb fusion for 3d human body reconstruction in all weather conditions." In 2023 IEEE International Conference on Robotics and Automation (ICRA), pp. 2752-2758. IEEE, 2023.
>
> [5] Zhang, Yiyuan, Kaixiong Gong, Kaipeng Zhang, Hongsheng Li, Yu Qiao, Wanli Ouyang, and Xiangyu Yue. "Meta-transformer: A unified framework for multimodal learning." arXiv preprint arXiv:2307.10802 (2023).

---

> > ### Comment · Reviewer_Rmx6 · 2024-11-24
> >
> > I appreciate the authors' response, which addresses most of my concerns.

---

> > > ### Comment · Reviewer_Rmx6 · 2024-11-26
> > >
> > > The authors conducted the required experiments by the reviewers. I raised the score. I believe the paper is worth acceptance in the current version.

---

> > > > ### Author Response · Authors · 2024-11-26
> > > > **Appreciation for the constructive comments**
> > > >
> > > > We sincerely appreciate your helpful suggestions! Thanks for your time and effort!

---

> ### Author Response · Authors · 2024-11-23
> **Looking forward to your reply**
>
> Dear Reviewer Rmx6,
>
> As the rebuttal deadline is approaching, could you please have a look at our response? We are looking forward to your reply!
>
> Feel free to let us know if you have any other concerns. Thanks for your time and effort!
>
> Best Regards,
>
> Authors of Submission 6113

---

> ### Author Response · Authors · 2024-11-24
> **Looking forward to your reply**
>
> Dear reviewer Rmx6,
>
> As the rebuttal deadline is approaching in **2 days**, could you please have a look at our response? We are looking forward to your reply!
>
> Feel free to let us know if you have any other concerns. Thanks for your time and effort!
>
> Best Regards,
>
> Authors of Submission 6113

---

> ### Author Response · Authors · 2024-11-25
> **Thanks and kind request for rate increment**
>
> Dear reviewer Rmx6,
>
> Thank you for acknowledging that our rebuttal has mostly addressed your initial concerns. We greatly appreciate your engagement and constructive comments.
>
> Currently three reviewers have responded that the rebuttal has addressed their concerns and may increase the score, (while two reviewers have not responded yet). It is believed that the revised manuscript has been **signficantly improved by address your concerns and adding many experimental results and analysis**. If so, we kindly request if you could increase the rating of our submission, marking the significance of our revisions according to your valuable suggestions.
>
> Thank you so much for your time and consideration.
>
> Best regards,
>
> Authors of Submission 6113

---

### Official Review · Reviewer_meFd · 2024-11-04

**Soundness:** 3
**Presentation:** 2
**Contribution:** 3
**Rating:** 6
**Confidence:** 3

**Summary:**

This paper introduces a model to perform human sensing on various input modalities and output tasks. The method utilizes transformer-based attention to input a varying number of modalities and learn complementary features. The method can perform during test time with any combination of input modalities better than a fusion architecture trained with those modalities, which indicates a step towards developing a multi-sensor foundation model for human motion understanding.


This review follows the sections of the paper.

**Strengths:**

Introduction:

1. The need for a flexible multimodal human sensing model is well-motivated. Particularly, the second paragraph outlining specific modalities and how their strengths and weaknesses can compensate for each other is strong.

Modality Invariant Foundation Model for Human Sensing:

2. The motivation for the cross-attention multimodal fusion (X-Fusion) is good. Attempting to combat the domination of certain modalities by incorporating modality-specific information is good. How did the authors know that the attention mechanism was weighing some modalities more than others? Did they view the transformers' attention weights and notice a trend? Did previous works report that happening? It would be great to see these preliminary results somewhere.
3. I appreciate the thought and structure that went into randomizing the modalities in "3.3 Modality-invariant training." Table 3 is good. How did you determine the final probabilities? How did you know it was necessary to include more LiDAR samples than the other modalities?

Experiments:

4. Just to clarify, in tables 1 and 2 X-Fi was only trained once and tested on various input modalities (each unique row), but the baselines were retrained on each modality combination separately. Is this true (line 365)? If this is the case, I think this should be noted in the caption and is a strength for your method.
5. I like Figure 4's clusterability scores. The visualization makes it hard to distinguish the difference between R and W+R, but I think the numbers emphasize the point that multimodal is better.

**Weaknesses:**

Introduction:

1. Figure 1: I am not sure what the purpose of this figure is. What are you trying to show? X-Fi can replace a bunch of modality-specific models? But within X-Fi we have all the modality-specific encoders anyways. Are you trying to indicate that you don't need to retrain models for each combination of modalities? Nothing in this figure indicates retraining. I would suggest brainstorming a more informative diagram.
2. Figure 2 is good, but could use some clarifications. Why are the last 3 modalities with dotted lines? Can you label the dark gray background boxes (which module is the X-Fusion part)? Also the text is quite small and hard to read. What does N stand for in the arrow loop?
3. I think the term "foundation model" is starting to be used loosely quite frequently, and I would appreciate if the authors could clarify in a sentence or two. Maybe define a foundation model in this context and how their method is such before referring to their method as a foundation model. This could also be done in related works, comparing your method to any related foundation models.

Related Works:

4. I think a discussion of multimodal missing modalities literature is missing. There are many works that perform multi-modal HAR in the face of missing modalities during training/testing which is a similar setup to X-Fi. Also, there is a lot of literature on transformer based fusion that could be mentioned here.
Here are some potentially relevant works:
- Sangmin Woo, Sumin Lee, Yeonju Park, Muhammad Adi Nugroho, and Changick Kim. Towards
Zihui Xue, Zhengqi Gao, Sucheng Ren, and Hang Zhao. The modality focusing hypothesis: Towards understanding cross modal knowledge distillation. arXiv preprint arXiv:2206.06487, 2022.
- Quan Kong, Ziming Wu, Ziwei Deng, Martin Klinkigt, Bin Tong, and Tomokazu Murakami. Mmact: A large-scale dataset for cross modal human action understanding. In Proceedings of the IEEE/CVF International Conference on Computer Vision, pp. 8658–8667, 2019.
- Qi Wang, Liang Zhan, Paul Thompson, and Jiayu Zhou. Multimodal learning with incomplete modalities by knowledge distillation. In Proceedings of the 26th ACM SIGKDD International Conference on Knowledge Discovery & Data Mining, pp. 1828–1838, 2020.
- XB Bruce, Yan Liu, and Keith CC Chan. Multimodal fusion via teacher-student network for indoor action recognition. In Proceedings of the AAAI Conference on Artificial Intelligence, volume 35(4), pp. 3199–3207, 2021.

5. In the second sentence under Modality invariant methods. I would disagree that decision-level approaches are restricted to fixed modality combinations, and both citations provided are feature level fusion. Decision level methods may average the outputs of separate modality modals, or perform some sort of voting, so it would be easy to extend to more modalities or drop the outputs of some modalities. In fact I think that's what your baseilne1 does to my understanding.
6. I don't think the discussion in the last paragraph citing LLM methods and a visual odometry method in robotics is very relevant.

Modality Invariant Foundation Model for Human Sensing:

7. A lot of notation is introduced and it is hard to follow. Consider maybe labeling the arrows and boxes in figure 2 with the notation ($G_{ca}, G_{cm}, Emb_{cm}$ etc.). Also maybe the dimensions of the cubicle blocks so we can see what n_f and d_f is referring to. Is $\theta_{Enc}$ Just the concatenation of $\theta_{E}$ and $\theta_{LP}$? Is positional encoding shown in Equation (1)?
8. The key value generators and cross-attention transformers is unclear. It would be helpful to provide a diagram (at least in the appendix) for them. Also, when using the term transformer, I assume the input is one set of tokens and that's projected into keys, queries and values, but "in cross-attention transformer," the input is already K,Q,V's. I would suggest referring to it as just a cross-attention module. The original Transformer in Vaswani et al (the only cited Transformer paper in 3.2.1) includes the KQV projection as well as an encoder with self-attention and a decoder with self-attention/cross attention and multiple heads. Is that what your "Transformer" is doing?
9. In line 272, I don't understand why Emb_{cm} is being added back to O_i ? Again a diagram here might help.
10. I would suggest an ablation without the K,V generators. It is just two MLP's but it happens right after the modality-specific linear projections, so maybe the modality-specific projections can capture that information.

Experiments:

11. In Tables 1 and 2 I would suggest labeling the reported metrics either in the column header or caption (it sounds like MSE for HPE and Accuracy for HAR?).
12. HPE Qualitative results Figure 3. It is not apparent to me that RGB captures better leg movements and depth is better for hands. Can you circle on the images where you are observing this? Also, any thoughts on why this would be the case? Maybe even providing numbers for the errors on hand joints vs leg joints could make this result more convincing!
13. The results could be strengthened by comparing to other multimodal methods. Are there other works that perform multimodal HPE or HAR on the MM_Fi and XRF55 datasets that you can compare to?

**Questions:**

I presented many questions and suggestions in the sections above. I think the main issue of the paper was a lack of clarity in describing the method and its implementation.

---

> ### Author Response · Authors · 2024-11-21
> **Response to Reviewer meFd (1/4)**
>
> We sincerely appreciate the reviewer meFd for the insightful questions and constructive comments. We are glad that the reviewer acknowledges that “**the need for a flexible human sensing model is well motivated**” and “**modality training is appreciated**”. Here we answer all the questions and add extensive experiments for clarification. We hope the responses can address the concerns. In addition, we have submitted a revised manuscript with an appendix where we mark all the suggested tables, figures, and analytics in magenta color.
>
> **Q1: Are there any previous works or experiment results that support the statement that the attention mechanism was weighing some modalities more than others?**
>
> **Answer**: Yes. [1] (CVPR 2022) reported the observation that ‘the Transformer models degrade dramatically with modal-incomplete data’. They trained a ViT on ‘full’ sets of image and text modalities and tested only with unimodal input. The result on MM-IMDb dataset reveals that the performance drops by 14.3% when missing image, and drops 36.7% when missing text. Similarly in [2] (CVPR 2023), the trained modality-invariant transformer model marginally depends on the presence of depth modality. ‘Dropping RGB barely decreases performance, while dropping Depth causes the localization to fail more drastically’, as they described.
>
> We admit that we didn’t clearly state the motivation of our X-Fusion design in section 3.2.2 and thus we modified the corresponding description. Additionally, we conducted a detailed literature review about the phenomenon that ‘the attention mechanism was weighing some modalities more than others’ and found the following works report the issue.
>
> **Q2: How did you determine the final probabilities of every modality in modality-invariant training procedures?**
>
> **Answer**: We started from setting equal probabilities for every modality, but then we found some modalities are under-fitted by the model. After evaluating the modality characteristics, we adjusted the corresponding modality probabilities to improve the performance.
>
> For the results presented in Table 3, we first evaluated the baseline results from the XRF55 dataset paper [3] and observed that the RFID modality performance is significantly worse than the other two, which motivated our first setting that emphasized RFID. However, after evaluating the outcomes with this setting, we noticed the model was underfitted on the WiFi modality. Consequently, we gradually increased the occurrence probability of WiFi to improve its performance.
>
> Similarly, for the MM-Fi dataset experiments, we first assessed the characteristics of all modalities and tested the model performance with equal modality occurrence probabilities (set to 0.5). Given the large performance gap between the baseline and model results for LiDAR, we decided to incorporate more LiDAR samples into the training process to address this disparity.
>
> **Q3: In Tables 1 and 2, was X-Fi trained once and tested on various modality combinations, while baselines were retrained for each combination?**
>
> **Answer**: Yes, X-Fi was only trained once with our proposed modality-invariant training strategy and tested on various modality combinations. And the baselines were trained on each combination separately, i.e. we trained 62 independent baseline models on MM-Fi dataset to form baseline results shown in Table 1. We emphasized the corresponding descriptions in section 4.1 and 4.2.
>
> **Q4:  Is Figure 1 trying to show that X-Fi can replace a bunch of modality-specific models and doesn't need to be retrained for each combination of modalities?**
>
> **Answer**: Yes, the purpose of Figure 1. is to show that X-Fi could provide one unified modality-invariant solution to all potential modality combinations. The left image shows that existing human sensing solutions are mostly designed on fixed modality combinations and we need to train different models with different modality combinations respectively for various application scenarios. In order to emphasize that X-Fi doesn’t need to be retrained for each combination, we modified the diagram correspondingly.
>
> **Q5: For Figure 2, why are the last 3 modalities with dotted lines? Can you label the dark gray background boxes (which module is the X-Fusion part)? Can you enlarge the font size in the diagram? What does $N$ stand for in the arrow loop?**
>
> **Answer**: The last 3 modalities with dotted lines represent those potential modalities that are not included in the given scenario, or we can call them ‘inactive’ modalities. However, those modalities could be utilized in other scenarios and the corresponding modality-specific structures will be activated
>
> We appreciate your suggestions on the diagram design and we labeled the dark gray background boxes, enlarged the text font size, and added more descriptions about the diagram to the caption in the paper.
>
> The $N$ in the arrow loop stands for the number of iterations on the X-Fusion block.

---

> ### Author Response · Authors · 2024-11-21
> **Response to Reviewer meFd (2/4)**
>
> **Q6: Why do you claim the approach is a ‘foundation model’ and how do you define it?**
>
> **Answer**: We appreciate the suggestion. X-Fi partially aligns with the foundation model concept by enabling modality-invariant representation learning. In current LLM research, a foundation model refers to a large-scale, pre-trained model that serves as a versatile base and could learn unified representations across diverse inputs [8][9]. They are capable of fine-tuning or adapting to a wide range of downstream tasks across domains.
>
> The foundation model we refer to is one that, given a set of modalities, the model requires only a single round of modality-invariant training and could handle versatile modality combinations during inference without the need for retraining or fine-tuning. X-Fi enables modality invariant representation learning, allowing it to process diverse multimodal combinations across different scenarios without the need for retraining on specific modality configurations, which partially aligns with the foundation model concept.
>
> However, we acknowledge that X-Fi has some limitations compared to existing LLM foundation models, as it requires larger-scale data from a wider range of scenarios for modality-invariant training to fully achieve the capabilities of a foundation model. We added foundation model related descriptions in the section 2.2 line 138-143 to clarify why our proposed X-Fi could be regarded as a foundation model.
>
> **Q7: Could the author add the discussion of multimodal missing modalities literature review?**
>
> **Answer**: We appreciate the suggestion and conducted literature reviews on the recommended papers. The discussion about these approaches and the corresponding citations are included in the related work section 2.2.
>
> **Q8: Are decision-level approaches restricted to fixed modality combinations? Since they can average or vote on outputs, they might easily handle adding or dropping modalities.**
>
> **Answer**: We agree with the reviewer and the decision-level fusion methods are not restricted to fixed modality combinations. We have added corresponding discussions in Section 2.2. However, the performance could be hindered by less informative modalities. For instance, due to the low resolution of WiFi, WiFi-based HPE results are worse than the other modalities and simply averaging the outputs of other modalities with WiFi can degrade the performance.
>
> **Q9: Why is the discussion in the last paragraph citing LLM methods and a visual odometry method in robotics very relevant?**
>
> **Answer**: This is because some novel multimodal LLM methods inspired our human sensing multimodal fusion design as they are proposed for cross modal fusion of image-text, embodied sensor-text, and RF sensors-text. Also, suggested by reviewer u1Rk , we cited some more relevant LLM works. The visual odometry method in robotics [2] introduced the concept of modality-invariant methods, which is one of our key concepts in the paper.
>
> **Q10: Could the author label the arrows and boxes in figure 2 with the notation? what does $n_{f}$ and $d_{f}$ refer to? Is  $\theta_{E}$ Just the concatenation of $\theta_{F}$ and $\theta_{LP}$ in equation(1)? Is positional encoding shown in Equation (1)?**
>
> **Answer**: We appreciate the suggestion. We have refined the notations and have added notations into Figure 2.
>
> The dimension of $n_{f}$ and $d_{f}$ we chose is 32 and 512, respectively, and we have added the description in section 4.1.
>
> The $\theta_{E}$ is the concatenation of $\theta_{F}$ and $\theta_{LP}$ since the feature encoders we designed consist of feature extractors and linear projectors. The positional encoding is not shown in Equation 1 because the equation is to describe the components within the feature encoders.
>
> **Q11: Should the author rename the “cross-attention transformer” to “cross-attention module”? Could the author provide detailed diagrams of the KV generators and the cross-attention modules?**
>
> **Answer**: We very much appreciate the suggestion and agree that “Cross-attention module” is indeed a better term to describe the designed cross-attention part. The cross-attention part we designed does not contain the QKV projection part and thus it couldn’t be named as “cross-attention transformer”. We revised all the “cross-attention transformer” terms to “cross-attention module” in the paper. We also added detailed diagrams of the KV generators and the cross-attention modules in the appendix Figure 7.

---

> ### Author Response · Authors · 2024-11-21
> **Response to Reviewer meFd (3/4)**
>
> **Q12: Why is the cross-modal embedding residual added back in the cross-modal transformer?**
>
> **Answer**: The cross-modal embedding is residual added back to the cross-modal transformer in order to retain cross-modal features. This residual connection is designed by considering that the cross attention fusion modules for each modality will cause the information loss of the previously learned cross-modal embedding, therefore downgrading the fusion efficiency. Such design will retain previously learned cross-modal features as much as possible while injecting the modality-specific information in the cross attention modules.
>
> **Q13: Why do we need k-v generators rather than directly use initial modality features as k-v?**
>
> **Answer**: As discussed in the section 3.2.2, the purpose of k-v generators is to preserve and refine the modality-specific information that will be injected back to the fusion process in the cross attention module. To prove that utilizing k-v generators to obtain key-value pairs is more efficient than directly using initial modality features as k-v, we conducted the ablation study of k-v generators on XRF55 HAR task in order to The results shown in the table below supports the efficiency of our k-v generator design and added to the appendix Table 9.
>
> | XRF55 | Without KV Generators | With KV Generators |
> |:---:|:---:|:---:|
> | R | 82.0% | **83.9%** |
> | W | 26.8% | **55.7%** |
> | RF | 40.0% | **42.5%** |
> | R+W | 85.0% | **88.2%** |
> | R+RF | 84.7% | **86.5%** |
> | W+RF | 50.2% | **58.1%** |
> | R+W+RF | 86.7% | **89.8%** |
>
> **Q14: In Tables 1 and 2 I would suggest labeling the reported metrics either in the column header or caption (it sounds like MSE for HPE and Accuracy for HAR?).**
>
> **Answer**: For HPE, MPJPE and PA-MPJPE are the metrics evaluated and Table 1. has listed the two metrics in the column header in the original manuscript. For HAR, we did miss the description of the metric utilized and we have added the ‘accuracy’ term in the caption to make clear of the metric.
>
> **Q15: Can you circle on the images where you are observing that RGB captures better leg movements and depth is better for the arm in Figure 3? Also, any thoughts on why this would be the case?**
>
> **Answer**: We zoomed in the areas where we observed that RGB captures better leg movements and depth is better for the arm, the picture after enlargement is added in the appendix Figure 5.
>
> To seek the reason for such observation, we explored the characteristics of the rgb data and the depth data in MM-Fi. We found that due to the arm's movement typically being faster than the legs, combined with the insufficient frame rate of the camera, the arms in the RGB samples are often blurry, while the legs are very clear. This results in poor RGB-based estimation performance for the arms. In contrast, in depth images, the arm contours are very clear, but the legs almost blend into the background, making their contours indistinct. Such inherent flaws in the data leads to poor depth-based estimation performance for the legs. We also included two images samples from rgb and depth in the appendix Figure 6.
>
> **Q16: Are there other works that perform multimodal HPE or HAR on the MM_Fi and XRF55 datasets that you can compare to?**
>
> **Answer**: We appreciate the suggestion and we went through all the papers citing MM-Fi and XRF55 datasets. To date, only one paper [4] used the MMFi and XRF55 datasets for multimodal HAR. But it focuses solely on modality alignment to improve single-modality results rather than conducting multi-modal fusion. Their model achieved 43.90% accuracy on the single LiDAR modality in the MM-Fi dataset, whereas our model reached 52.7%. On the XRF55 dataset's mmWave modality, their approach achieved 50.30%, while ours reached 83.9%.
>
> |  | BABEL [4] | X-Fi | Imp. $\uparrow$ |
> |---|---|---|---|
> | LiDAR (MM-Fi [5]) | 43.90% | **52.7%** | 8.8% |
> | mmWave (XRF55 [3]) | 50.30% | **83.9%** | 33.6% |
>
> Beside the research works based on the MM-Fi and XRF55 datasets, we also actively looked for multimodal fusion methods and carried out experiments on the dataset in order to strengthen our X-Fi results. The comparisons are added in the appendix Table 5 and shown below.
>
> | Modalities | Methods | MPJPE (mm) | PA-MPJPE (mm) |
> |:---:|:---:|:---:|:---:|
> | I + D + L | Meta-Transformer-B16F [7] | 105.2 | 70.3 |
> |  | Meta-Transformer-B16T [7] | 96.2 | 68.1 |
> |  | X-Fusion  | **84.8** | **48.2** |
> | I + D+ R  | ImmFusion [6] | 207.3 | 146.8 |
> |  | X-Fusion | **83.4** | **47.3** |
> | I + D + L + R + W | ImmFusion [6] | 107.9 | 63.7 |
> |  | Meta-Transformer-B16T [7] | 91.4 | 64.9 |
> |  | X-Fusion | **83.7** | **47.6** |

---

> ### Author Response · Authors · 2024-11-21
> **Response to Reviewer meFd (4/4)**
>
> **Reference:**
>
> [1] Ma, Mengmeng, Jian Ren, Long Zhao, Davide Testuggine, and Xi Peng. "Are multimodal transformers robust to missing modality?." In Proceedings of the IEEE/CVF Conference on Computer Vision and Pattern Recognition, pp. 18177-18186. 2022.
>
> [2] Memmel, Marius, Roman Bachmann, and Amir Zamir. "Modality-invariant Visual Odometry for Embodied Vision." In Proceedings of the IEEE/CVF Conference on Computer Vision and Pattern Recognition, pp. 21549-21559. 2023.
>
> [3] Wang, Fei, Yizhe Lv, Mengdie Zhu, Han Ding, and Jinsong Han. "XRF55: A Radio Frequency Dataset for Human Indoor Action Analysis." Proceedings of the ACM on Interactive, Mobile, Wearable and Ubiquitous Technologies 8, no. 1 (2024): 1-34.
>
> [4] Dai, Shenghong, Shiqi Jiang, Yifan Yang, Ting Cao, Mo Li, Suman Banerjee, and Lili Qiu. "Advancing Multi-Modal Sensing Through Expandable Modality Alignment." arXiv preprint arXiv:2407.17777 (2024).
>
> [5] Yang, Jianfei, He Huang, Yunjiao Zhou, Xinyan Chen, Yuecong Xu, Shenghai Yuan, Han Zou, Chris Xiaoxuan Lu, and Lihua Xie. "Mm-fi: Multi-modal non-intrusive 4d human dataset for versatile wireless sensing." Advances in Neural Information Processing Systems 36 (2024).
>
> [6] Chen, Anjun, Xiangyu Wang, Kun Shi, Shaohao Zhu, Bin Fang, Yingfeng Chen, Jiming Chen, Yuchi Huo, and Qi Ye. "Immfusion: Robust mmwave-rgb fusion for 3d human body reconstruction in all weather conditions." In 2023 IEEE International Conference on Robotics and Automation (ICRA), pp. 2752-2758. IEEE, 2023.
>
> [7] Zhang, Yiyuan, Kaixiong Gong, Kaipeng Zhang, Hongsheng Li, Yu Qiao, Wanli Ouyang, and Xiangyu Yue. "Meta-transformer: A unified framework for multimodal learning." arXiv preprint arXiv:2307.10802 (2023).
>
> [8] Jacob Devlin. Bert: Pre-training of deep bidirectional transformers for language understanding.
> arXiv preprint arXiv:1810.04805, 2018.
>
> [9] Hugo Touvron, Thibaut Lavril, Gautier Izacard, Xavier Martinet, Marie-Anne Lachaux, Timoth´ee
> Lacroix, Baptiste Rozi`ere, Naman Goyal, Eric Hambro, Faisal Azhar, et al. Llama: Open and
> efficient foundation language models. arXiv preprint arXiv:2302.13971, 2023

---

> ### Author Response · Authors · 2024-11-25
> **Looking forward to your reply**
>
> Dear reviewer meFd,
>
> As the rebuttal deadline is approaching in **2 days**, could you please have a look at our response? We are looking forward to your reply!
>
> Feel free to let us know if you have any other concerns. Thanks for your time and effort!
>
> Best Regards,
>
> Authors of Submission 6113

---

> ### Author Response · Authors · 2024-11-26
> **Looking forward to your reply**
>
> Dear reviewer meFd,
>
> As the rebuttal deadline approaches in **less than a day**, we kindly ask if you could take a moment to review our response. We are encouraged that four reviewers have acknowledged that our responses address most of their concerns, with three either raising or expressing their intention to raise their ratings.
>
> We are sincerely looking forward to your reply and feel free to let us know if you have any other concerns. Thanks for your time and effort!
>
> Best Regards,
>
> Authors of Submission 6113

---

> ### Comment · Reviewer_meFd · 2024-11-27
>
> I'm impressed with the author's rebuttal. I think they addressed most of my concerns well. However, I'm still wary of the strength of the overall work. In particular, addressing missing modalities is not very novel, and although other works may not use the exact terminology "modality-invariant" model, I do believe this has been investigated before in many applications, also using attention mechanisms. I am also still not convinced this model provides the sort of general reasoning a foundation model is supposed to represent and still believe some of the notation/formulation presented is slightly difficult to follow.
>
> Nonetheless, I think this iterative application of the cross-modal transformer with modality-specific and fused info is unique and interesting. To my understanding, this work's contribution is sufficient and relevant to ICLR. Thus, I will increase my score to marginally above acceptance.

---

> > ### Author Response · Authors · 2024-11-27
> > **Appreciation for the constructive comments**
> >
> > We sincerely appreciate your helpful suggestions! Thanks for your time and effort!

---

### Official Review · Reviewer_H5r3 · 2024-11-09

**Soundness:** 3
**Presentation:** 2
**Contribution:** 2
**Rating:** 6
**Confidence:** 4

**Summary:**

The paper presents X-FI, a foundation model for multi-modal human sensing. X-FI allows inference with versatile combinations of sensor modalities, and supports multiple tasks including pose estimation and activity recognition. Key to X-FI is a multi-modal feature encoding module that builds on cross-attention. X-FI was evaluated on two public datasets, demonstrating some encouraging results.

**Strengths:**

The paper addresses an interesting and important problem of developing foundation models for multi-modal human sensing.

There are some interesting ideas in the model design, e.g., using cross-attention to iteratively refine the average pooled multimodal features.

**Weaknesses:**

I am not convinced by the claim on “modality invariant human sensing”, one of the key contributions of the paper. Conceptually, there are several approaches for modality invariant processing. For example, the model in Memmel et al., 2023 (a Transformer that takes tokens from any combination of modalities) can in theory be adapted for human sensing. Technically, this invariance is achieved by dropping out modalities during training, which is again discussed in Memmel et al., 2023 and also mentioned Chen et al., 2023.

The experiments miss important baselines. There is hardly a comparison to prior methods in the paper. At least, two sets of baselines should have been considered: (a) latest methods using a single modality (to demonstrate the benefit of multi-modal fusion); and (b) previous methods that also considers multimodal fusion such as Chen et al., 2023 (to highlight the benefit of the proposed method). The current baselines are rather straw man.

The presentation of the technical components lacks clarity. Some of the designs are not well justified.
* Eq 2: The use of average pooling is not justified. In fact, with different numbers of input modalities, each bin in average pooling may span (1) tokens from a single modality; or (2) tokens from more than one modality. It is not clear why average pooling works here. One possibility is learning a fixed number of queries and using cross-attention to project the input multi-modal feature, similar to the design of Percerver IO.
* Eq 4: It is unclear how to determine the number of iterations for the X-fusion block. I also could not find the exact number used in the experiments.
* L229: I assume n_f is the number of tokens per modality, but it was not described in the paper.

================================

Post-rebuttal update: In the rebuttal, the authors have addressed my previous concerns, included substantial updated results, and majorly improved the paper. While I still have some concerns over the technical innovation and the use of average pooling, I feel like the paper is now beyond the bar. Therefore, I have increased my rating.

**Questions:**

Can you clarify the number of iterations needed for the X-fusion block?

It will be interesting to compare average pooling vs. cross-attention with learned queries.

Comparison to prior methods (see weaknesses) should be included.

---

> ### Author Response · Authors · 2024-11-21
> **Response to Reviewer H5r3 (1/3)**
>
> We appreciate the reviewer H5r3 for the constructive comments. We are glad that the reviewer acknowledges "**the paper addresses an interesting and important problem**". Based on the suggestions, we mainly address the lack of comparisons to prior methods and add experiments on other SOTA multimodal fusion methods to prove the significance of our proposed X-Fi. In addition, we have submitted a revised manuscript with an appendix in which we add all the suggested tables, figures, and analytics in magenta color.
>
> **Q1: Why do we claim modality invariant human sensing as one of our key contributions, since conceptually approaches proposed in [1] and [2] could in theory be adapted? Comparisons with prior multi-modal fusion approaches like [2] should be considered to better highlight the strengths of the proposed method?**
>
> **Answer**: We claim it as one of our key contributions since *modality-invariant human sensing* is a new task in human perception, i.e., human activity recognition (HAR) and human pose estimation (HPE). As discussed in the manuscript, though some previous works also studied modality-invariant [1] or multimodal fusion [2] tasks, they differ significantly from ours regarding the task, model architecture, training strategy, and experiments. In this answer, we support our argument with both conceptual comparison and supplementary experiments. We first compare the conceptual difference between our work and [1][2] in the table below.
>
> |  | Ours | ImmFusion [2] | VO Transformer [1] |
> |---|---|---|---|
> | Task | Modality-invariant human perception (flexible combinations of modalities for HAR and HPE) | mmWave-RGB fusion for human body reconstruction  (fixed two modalities for dense HPE) | Modality-invariant visual odometry  (two vision modalities only) |
> | Model Architecture | X-Fi consists of a cross-modal transformer to handle dynamic modality combinations and learn unified cross-modal features, and a cross-attention fusion modules that retain modality-specific features. X-Fi allows more modalities in a flexible input manner. | ImmFusion employs a transformer module to fuse the concatenated point cloud features and image features. It only allows two modalities and fixed modalities. | VO Transformer uses a vanilla transformer encoder to perform modality-invariant fusion. |
> | Training Strategy | The proposed modality invariant training strategy uses a modality existence list to adjust independent occurrence probability for each modality. Suitable for datasets with many modality combinations, e.g., 31 combinations in MM-FI [3]. | In the training process, the modality mask is used to partially mask a portion of every image sample to simulate sensor distortion. No entire modality dropping applied. | The proposed modality invariant training strategy sets sampling percentages for each modality combination, which is impractical for a large number of modality combinations. |
> | Experiments | The experiments are conducted on 2 large humans sensing datasets with different 5 modalities that cover both vision-based and radio-frequency-based sensors.  The model is trained once and validated on **all 31 modality combinations**. | The experiments are only conducted on RGB and mmWave modalities. The model is trained and evaluated on a **fixed modality combination**. | The experiments are  validating on 2 vision-based modalities. The model is trained once and evaluated on **3 possible modality combinations**. |
>
> In summary, our proposed X-Fi (1) deals with a more difficult task with 5 different modalities and flexible inputs, (2) proposes a novel structure that allows flexible inputs with possibly missing modalities (instead of vanilla transformer), (3) presents a novel training strategy that drops whole modalities for training, and (4) conducts extensive experiments on 2 datasets with 31 modalities combinations, demonstrating the significant improvement.
>
> As suggested, to better demonstrate the significance of our proposed X-Fi, we expand the ImmFusion method by concatenating multimodal features and expand the VO Transformer method by feeding all modalities’ features into a single transformer structure, which are then compared to our method on the benchmark.
>
> For ImmFusion [2] on HPE task, we compare it on the original setting (image+point cloud, I+D+R), and on all modality settings (I+D+R+L+W). The results are shown below and indicate the superior performance of our proposed X-Fi.
>
> | MM-Fi HPE | MPJPE (mm) |  | PA-MPJPE (mm) |  |
> |:---:|:---:|:---:|:---:|:---:|
> | Method | ImmFusion [2] | X-Fi | ImmFusion [2] | X-Fi |
> | I + D + R | 207.3 | **83.4** | 146.8 | **47.3** |
> | I + D + L + R + W | 107.9 | **83.7** | 63.7 | **47.6** |

---

> ### Author Response · Authors · 2024-11-21
> **Response to Reviewer H5r3 (2/3)**
>
> **Q1: Why do we claim modality invariant human sensing as one of our key contributions, since conceptually approaches proposed in [1] and [2] could in theory be adapted? Comparisons with prior multi-modal fusion approaches like [2] should be considered to better highlight the strengths of the proposed method?**
>
> **Answer**: For VO Transformer [1], we have already compared our method with VO Transformer in the original manuscript Section 4.3. The transformer based fusion method is the VO Transformer structure, and we have added the references here for better clarification. The experiment results have been listed in Table 4 and Table 5.
>
> | MM-Fi HPE | MPJPE (mm) |  | PA-MPJPE (mm) |  |
> |:---:|:---:|:---:|:---:|:---:|
> | Method | VO Transformer [1] | X-Fi | VO Transformer [1] | X-Fi |
> | I | 99.6 | **93.9** | 65.0 | **60.3** |
> | L | 212.2 | **167.1** | 109.6 | **103.2** |
> | R | 138.0 | **127.4** | 74.6 | **69.8** |
> | W | 243.3 | **225.6** | 112.9 | **105.3** |
> | L + R | 123.4 | **109.8** | 69.9 | **63.4** |
> | L + W | 216.3 | **159.5** | 112.7 | **102.7** |
> | I + L + W | 96.6 | **93.0** | 61.5 | **59.7** |
>
> Additionally, we also added the comparison with another SOTA multimodal fusion method Meta-Transformer [4], which is trained and evaluated on fixed modality combinations. We used the Meta-Transformer-B16 encoder as the backbone to carry out experiments. The results shown below demonstrate X-Fi’s superior performance.
>
> | MM-Fi HPE | MPJPE (mm) |  |  | PA-MPJPE (mm) |  |  |
> |:---:|:---:|:---:|:---:|:---:|:---:|:---:|
> | Method | Meta Transformer-B16F [4] | Meta Transformer-B16T [4] | X-Fusion | Meta Transformer-B16F [4] | Meta Transformer-B16T [4] | X-Fusion |
> | I + D + L | 105.2 | 96.2 | **84.8** | 70.3 | 68.1 | **48.2** |
> | I+D+L+R+W | - | 91.4 | **83.7** | - | 64.9 | **47.6** |
>
> **Q2: Are there any additional baselines, such as comparing recent single-modality methods to demonstrate the advantages of multi-modal fusion. The experiments miss important baselines.**
>
> **Answer**: As discussed in Table 1 and 2 of the original manuscript, we have already included results of the single modality with latest methods as the backbone. Moreover, the detailed description of the selected single modality backbones are illustrated in appendix A.1. The comparisons between single modality results and X-Fi multimodal fusion results demonstrate the superior multimodal fusion ability of the proposed Xi-Fi. For example, the image plus mmWave radar multimodal result shown in line 331 surpasses both the image and mmWave radar single-modal baseline results in line 325 and line 328 by 26.6% and 37.2% in terms of MPJPE, respectively. Due to page limitations, we had to put the settings and detailed descriptions in the appendix Section A.1, which may have led to misunderstandings. We revised and emphasized corresponding descriptions in the main text.
>
> **Q3: What is the number of iterations needed for the X-fusion block in experiments? How to determine the number of iterations for the X-fusion block?**
>
> **Answer**: The default iteration is 4 as illustrated in our experiment setting. As suggested, we added the missing descriptions about the number of iterations for the X-fusion block to the paper in line 381. To illustrate how the number of iterations affects the model performance and to figure out how to determine the number, we carry out a new experiment with different the number of iterations on XRF55 dataset for HAR task. The result and the corresponding discussion shown below are added into the appendix A.4 and Table 8.
>
> | XRF55 | number of iteration = 2 | number of iteration = 4 | number of iteration = 5 |
> |---|---|---|---|
> | R | 80.6% | **83.9%** | 82.6% |
> | W | 24.2% | **55.7%** | 55.4% |
> | RF | 38.8% | **42.5%** | 39.1% |
> | R+W | 84.1% | 88.2% | **88.7%** |
> | R+RF | 84.1% | **86.5%** | 84.8% |
> | W+RF | 48.0% | 58.1% | **58.7%** |
> | R+W+RF | 86.3% | **89.8%** | 89.1% |
>
> When the number of iterations is set to 2, the model still cannot generate good results on  some hard modalities such as WiFi and RF. In these modalities, the resolution is low so it’s harder to learn modality-specific features. To further explore if the default 4 iterations are sufficient, we increased the number of iterations to 5. However, the model can only generate similar results as the iteration to be 4. In summary, we currently set the iteration to be 4 which has been sufficient, but we believe precisely tuning this hyperparameter can further achieve better results.

---

> ### Author Response · Authors · 2024-11-21
> **Response to Reviewer H5r3 (3/3)**
>
> **Q4: It will be interesting to compare average pooling vs. cross-attention with learned queries for dimension reduction. Need to justify why average pooling is designed in the cross-modal transformer.**
>
> **Answer**: It is a quite interesting idea and we added an experiment. We do realize the cross-attention with learned queries might be a good solution for dimension reduction, which have recently been proved to be effective in the Vision-Language Model paper (Q-Former) [5] and vision soft prompt [6]. To this end, we conducted comparison experiments by replacing the average pooling based dimension reduction with the cross-attention based dimension reduction.
> However, we find it hard to make the cross-attention based dimension reduction converge during modality invariant multimodal training this week. The training process is illustrated in appendix Figure 8.
>
> Possible reasons are (1) our datasets include a large amount of modality pairs, making it hard to utilize the existing feature encoders to achieve modality alignment, (2) the scale of multimodal data pairs is not as large as that of image-text pairs, making it hard to conduct large-scale encoder pretraining. Thus we keep average pooling in our current design, which is a simple solution for dimension reduction for now.
>
> Utilizing cross-attention with learned queries for dimension reduction is a relatively independent novelty for multimodal fusion research. In the future, we will adopt the two-stage training strategy in vision-language models, starting with large-scale pre-training to develop strong encoders for tight correlation among modality features, followed by training the subsequent components that include utilizing cross-attention for dimensionality reduction.
>
> **Q5: What is the definition of $n_{f}$ in L229 and what is the number used in the experiment?**
>
> **Answer**: The $n_{f}$ represents the number of feature tokens for each modality feature and the number used in the experiment is 32. We have added the missing descriptions of $n_f$ in line 161 and 375.
>
> **Reference**:
>
> [1] Memmel, Marius, Roman Bachmann, and Amir Zamir. "Modality-invariant Visual Odometry for Embodied Vision." In Proceedings of the IEEE/CVF Conference on Computer Vision and Pattern Recognition, pp. 21549-21559. 2023.
>
> [2] Chen, Anjun, Xiangyu Wang, Kun Shi, Shaohao Zhu, Bin Fang, Yingfeng Chen, Jiming Chen, Yuchi Huo, and Qi Ye. "Immfusion: Robust mmwave-rgb fusion for 3d human body reconstruction in all weather conditions." In 2023 IEEE International Conference on Robotics and Automation (ICRA), pp. 2752-2758. IEEE, 2023.
>
> [3] Yang, Jianfei, He Huang, Yunjiao Zhou, Xinyan Chen, Yuecong Xu, Shenghai Yuan, Han Zou, Chris Xiaoxuan Lu, and Lihua Xie. "Mm-fi: Multi-modal non-intrusive 4d human dataset for versatile wireless sensing." Advances in Neural Information Processing Systems 36 (2024).
>
> [4] Zhang, Yiyuan, Kaixiong Gong, Kaipeng Zhang, Hongsheng Li, Yu Qiao, Wanli Ouyang, and Xiangyu Yue. "Meta-transformer: A unified framework for multimodal learning." arXiv preprint arXiv:2307.10802 (2023).
>
> [5] Li, Junnan, Dongxu Li, Silvio Savarese, and Steven Hoi. "Blip-2: Bootstrapping language-image pre-training with frozen image encoders and large language models." In International conference on machine learning, pp. 19730-19742. PMLR, 2023.
>
> [6] Zhou, Kaiyang, Jingkang Yang, Chen Change Loy, and Ziwei Liu. "Learning to prompt for vision-language models." International Journal of Computer Vision 130, no. 9 (2022): 2337-2348.

---

> > ### Author Response · Authors · 2024-11-24
> > **Looking forward to your reply**
> >
> > Dear reviewer H5r3,
> >
> > As the rebuttal deadline is approaching in **2 days**, could you please have a look at our response? We are looking forward to your reply!
> >
> > Feel free to let us know if you have any other concerns. Thanks for your time and effort!
> >
> > Best Regards,
> >
> > Authors of Submission 6113

---

### Author Response · Authors · 2024-11-21
**Summary of the rebuttal and the major changes of revised manuscript**

Dear reviewers,

We would like to express our heartfelt gratitude for your invaluable time, expertise, and meticulous attention in reviewing our manuscript. The insightful comments and constructive feedback have immensely enriched the quality and rigor of our work.

We appreciate that all the reviewers acknowledge the advantages of our work: “**the paper addresses an interesting and important problem**” (Reviewer H5r3), “**the model is well-motivated**” (Reviewer meFd), “**the paper presents a simple yet effective framework**” (Reviewer Rmx6), “**the paper addresses an important problem and is clearly written and well-structured**” (Reviewer u1Rk), “**the author made a meaningful attempt and the model is a promising approach**”(Reviewer rmbV).

We also realized that some concerns should be addressed to improve our manuscript. Hence we have diligently addressed all inquiries by conducting extensive experiments to support our responses. Furthermore, **a revised manuscript** has been submitted, accompanied by an appendix that delineates all revisions, highlighted in *magenta color* for clarity. Owing to space constraints, selected experiments have been incorporated into the main manuscript while most of the supplementary experiments have been included in the appendix. Allow me to summarize the significant alterations made in both the rebuttal and the revised manuscript:

**Expanded Baselines**: Included two advanced multimodal fusion and human sensing methods (ImmFusion, Meta-Transformer) as baselines to demonstrate the effectiveness of the proposed method in modality-invariant human sensing.

**Ablation Study Enrichment**: Expanded 4 new ablation studies including experiments and analysis of Key-Value generators, the number of iterations on X-Fusion block, the number of stacked blocks in X-Fusion and the number of layers in cross-modal transformer.

**Exploring 3 Interesting Ideas Raised by Reviewers**: Conducted experiments based on the suggestions about replacing the average pooling based dimension reduction with the **cross-attention based dimension reduction**, adding **contrastive loss between modality-specific features and cross-modal embedding**, and conducting continual learning experiment by **gradually adding new modality each time**.

**Notation Revision and Explanation**: Revised all the notation errors and provided clear descriptions. Added clear notations to the Figure 2 to make them easy to follow.

**Figure Revision and Enhancement**:  2 Figures are revised and 3 Figures are added. In detail, revised Figure 1 to clarify the purpose of the conceptual diagram. Revised Figure 2 to clearly indicate each part of the diagram and provided the detailed diagram of key-value generators and cross-attention modules in appendix Figure 7. Added detailed comparison of HPE qualitative results in Figure 3 to appendix, Figure 5 and Figure 6 for more intuitive illustration.

**Literature Review Enrichment**: Added the discussion of the definition of ‘foundation model’ in our paper. Enriched the citations regarding multimodal and modality invariant methods.

**Supplementary model parameter calculation**: Added the comparison of the number of parameters in the VO Transformer and X-Fusion block for fair performance comparison.

We believe that the revisions made to the manuscript, along with the detailed rebuttals, have significantly improved its quality and addressed the concerns raised, making it more satisfactory for publication.

Best Regards,

Authors of Submission 6113

---

### Meta-Review · Area_Chair_DD3k · 2024-12-16

**Metareview:**

This paper proposes X-Fi, a multi-modal foundation model designed for human sensing, capable of integrating diverse sensor modalities for tasks such as human pose estimation and activity recognition. Utilising a transformer-based architecture with the proposed X-fusion mechanism, X-Fi preserves modality-specific features during fusion, enabling flexibility in handling varying modality combinations without retraining. The model was evaluated on benchmark datasets, showing improved performance over existing methods in both pose estimation and activity recognition tasks. The paper's strengths include its innovative use of cross-attention for modality-specific feature fusion, the flexibility of its modality-invariant design, and effective experimentation with multiple modality combinations. The clear motivation, robust model design, and well-explained methodology contribute to its impact on multi-modal human sensing. As a result, all reviewers have expressed positive feedback and recommended "accept" or "marginally above the acceptance threshold".

**Additional Comments On Reviewer Discussion:**

Reviewers highlighted several limitations, including missing important baselines in the experiments and a lack of clarity in the presentation of technical components, such as the use of average pooling and iteration details for the X-fusion block. They also noted the limited evaluation on datasets with specific modalities. Additionally, the absence of ablation studies on the cross-attention module and its efficacy in extracting modality-specific features was pointed out, along with the need to explore how varying transformer layers for different modality counts might impact results. The rebuttal addressed these major concerns, and all reviewers expressed satisfaction with the responses.

---

### Decision · Program_Chairs · 2025-01-22

Accept (Poster)